# Sampling is as easy as keeping the consistency: convergence guarantee for Consistency Models

## Abstract

We provide the first convergence guarantee for the Consistency Models (CMs), a newly emerging type of one-step generative models that is capable of generating comparable samples to those sampled from state-of-the-art Diffusion Models. Our main result is that, under the basic assumptions on score-matching errors, consistency errors, and smoothness of the data distribution, CMs can efficiently generate samples in one step with small $W_2$ error to any real data distribution. Our results (1) hold for $L^2$-accurate assumptions on both score and consistency functions (rather than $L^\infty$-accurate assumptions); (2) do not require strong assumptions on the data distribution such as log-Sobelev conditions; (3) scale polynomially in all parameters; and (4) match the state-of-the-art convergence guarantee for score-based generative models. We also show that the Multi-step Consistency Sampling procedure can further reduce the error comparing to one step sampling, which supports the original statement of Song et al. (2023). Our result can be generalized to arbitrary bounded data distributions that may be supported on some low-dimensional sub-manifolds. Our results further imply TV error guarantees when making some Langevin-based modifications to the output distributions.

## 1 Introduction

Score-based generative models (SGMs), also known as diffusion models (Sohl-Dickstein et al. (2015); Song & Ermon (2019); Dhariwal & Nichol (2021); Song et al. (2021; 2020)), are a family of generative models which achieve unprecedented success across multiple fields like image generation (Dhariwal & Nichol (2021); Nichol et al. (2021); Ramesh et al. (2022); Saharia et al. (2022)), audio synthesis (Kong et al. (2020); Chen et al. (2020); Popov et al. (2021)) and video generation (Ho et al. (2022a;b)); see, e.g., the recent surveys (Cao et al. (2022); Croitoru et al. (2022); Yang et al. (2022)). A key point of diffusion models is the iterative sampling process which gradually reduces noise from random initial vectors which provides a flexible trade-off of compute and sample quality, as using extra compute for more iterations usually yields samples of better quality. However, compared to single-step generative models like GANs, VAEs, or normalizing flows, the generation process of diffusion models requires 10-2000 times more, limiting it to a small number of real-time applications.

To overcome this limitation, Song et al. (2023) proposed Consistency Models (CMs) that can directly map noise to data, which can be seen as an extension of SGMs. CMs support fast one-step generation by design, while still allowing multistep sampling to trade compute for sampling quality. CMs can be trained either by distilling pre-trained diffusion models or as stand-alone generative models altogether. Song et al. (2023) demonstrates its superiority through extensive experiments outperforming existing distillation techniques for diffusion models, and when trained in isolation, CMs outperform existing one-step, non-adversarial generative models on standard benchmarks.

Besides the achievements of CMs in saving the the generation costs as well as keeping the sampling quality, it is a pressing question of both practical and theoretical concern to understand the mathematical underpinnings that explain their startling successes. The theoretical guarantee for SGMs has been extensively studied and well established Chen et al. (2022); Lee et al. (2022a;b). Despite the

theoretical successes of SGMs, one would wonder if CMs can inherit the good points from SGMs, as they are inextricably linked in their underlying mathematical process.

Providing a convergence analysis for CMs and exploring the benefits for CMs compared to SGMs is a pressing first step towards theoretically understanding why CMs work in practice.

## 1.1 OUR CONTRIBUTIONS

In this work, we take a step towards connecting theory and practice by providing a convergence guarantee for CMs, under minimal assumptions that coincide with our intrinsic. For the underlying SGMs, we make no more assumptions than the state-of-the-art works:

**A1** The score function of the forward process is $L_s$-Lipschitz.
**A2** The second moment of the data distribution $p_{\text{data}}$ is bounded.

Note that these two assumptions are standard, no more than what is needed in prior works. The crucial point to these two assumptions is that they do not need log-concavity, a log-Sobelev inequality, or dissipativity, which cover arbitrarily non-log-concave data distributions. Our main result is summarized informally as follows.

**Theorem 1.** *Under Assumptions* **A1** *and* **A2***, and in addition if the consistency model is Lipschitz and the consistency error, score estimation error in $L^2$ are at most $O(\varepsilon)$ with an appropriate choice of step size in the training procedure, then, the CM outputs a measure which is $\varepsilon$-close in Wasserstein-2 ($W_2$) distance to $p_{\text{data}}$ in single step.*

We find Theorem 1 surprising, because it shows that CMs can output a distribution arbitrary close to the data distribution in $W_2$ distance with a single step. The error of CMs is just the same order as what SGMs achieved, under the assumption that the consistency error is small enough, which coincides with the incredible success of CMs in many benchmarks. In the fields of neural networks, our result implies that so long as the neural network succeeds at the score and consistency function estimation tasks, the remaining part of the CM algorithm is almost understood, as it admits a strong theoretical justification.

However, learning the score function and consistency function is also difficult in general. Nevertheless, our result still leads the way to further investigations, such as: do consistency function for real-life data have intrinsic structure which can be well-explored by neural networks? If the answer is true, this would then provide an end-to-end guarantee for CMs.

**Better error bounds by multistep consistency sampling.** Beyond the one step CM sampling, Song et al. (2023) suggests a way for multistep sampling to trade compute for sampling quality. However in their original work, they did not make any analysis on the positive effect of multistep sampling comparing to one-step sampling. In our work, we analysis the error bound for each middle state of multistep sampling, showing an asymptotically convergent error bound that is greatly smaller than the error bound for one-step sampling. Our analysis reveals the fact that with a suitable choice of middle time points, the multistep consistency sampling takes a few more steps to achieve the near-best performance.

**Bounding the $W_2$ error for general bounded data distribution.** The foregoing results are established on the assumption **A1**, which only holds for $L_s$-smooth data distribution. When only assuming the data distribution to be bounded supported, which includes the situation when $q$ is supported on a lower-dimensional submanifold of $\mathbb{R}^d$, we can still guarantee polynomial convergence in the Wasserstein metric by early stopping. As the methodology is the same as in Chen et al. (2022) and Lee et al. (2022b), we do not claim the originality, but just include this part for completeness.

**Bounding the Total Variational error.** As the mathematical foundation of CMs established on the reverse probability flow ODE, they share the same shortcoming compared to the probability flow SDE: they can not get an error bound in Total Variational (TV) distance or Kullback-Leibler divergence by only controlling the score-matching objective, thus may even fail to estimate the likelihood of very simple data distributions (Fig.1 in Lu et al. (2022)). To solve this potential problem, we offered two modification processes to control the TV error: we can take an OU-type smoothing which takes no more evaluation costs to get a relatively larger TV error bound; we can also apply a Langevin dynamics for correcting purpose with the score model to get a smaller TV error bound, while it needs additional $O(\varepsilon^{-1}d^{1/2})$ evaluation steps.

## 1.2 PRIOR WORKS

As far as we know, this is the first work to establish a systematical analysis of the convergence property of CMs. As the CMs and SGMs share a similar mathematical essence in the asymptotic situation, our result can be compared to a vast list of literature on the convergence of SGMs.

**SDE-type SGMs.** The Langevin Monte Carlo (LMC) algorithm (Rossky et al. (1978)) can be seen as the predecessor to the SDE-type SGMs, and literature on the convergence of LMCs is extensive, such as Durmus & Moulines (2015); Cheng & Bartlett (2017); Cheng et al. (2017). However, these works mainly consider the case of exact or stochastic gradients. By the structure of the score-matching loss function, only an $L^2$-accurate gradient can be guaranteed for SDE-type SGMs. [Lee et al. (2022a)] is the first to give a polynomial convergence guarantee in TV distance under an $L^2$-accurate score. However, they rely on the data distribution satisfying smoothness conditions and a log-Sobolev inequality, which essentially limits the guarantees to unimodal distributions.

Bortoli (2022) instead only make minimal data assumptions, giving convergence in Wasserstein distance for distributions with bounded support $\mathcal{M}$. In particular, this covers the case of distributions supported on lower-dimensional manifolds, where guarantees in TV distance are unattainable. However, their guarantees have exponential dependence on the diameter of $\mathcal{M}$ or other parameters such as the Lipstchitz constant of score function.

Recently, Chen et al. (2022) and Lee et al. (2022b) concurrently obtained theoretical guarantees for SGMs under similar general assumptions on the data distribution. They give Wasserstein bounds for any distribution of bounded support (or sufficiently decaying tails), and TV bounds for distributions under minimal smoothness assumptions, that are polynomial in all parameters. This gives theoretical grounding to the success of SGM of data distribution that is often non-smooth and multimodal.

**ODE-type SGMs.** Instead of implementing the time-reversed diffusion as an SDE, it is also possible to implement it as an ordinary differential equation (ODE). However, current analyses of SGMs cannot provide a polynomial-complexity TV bound of the probability flow ODE under minimal assumption on data distribution. Lu et al. (2022) first bounded the KL divergence gap (and thus TV error) by higher-order gradients of the score function, and thus suggested controlling this bound by minimizing the higher order score-matching objectives, which causes much more difficulties in training the score model.

Instead of changing the training procedure, Chen et al. (2023b) obtained a discretization analysis for the probability flow ODE in KL divergence, though their bounds have a large dependency on $d$, exponential in the Lipschitz constant of the score integrated over time, which rely on higher order regularities of the log-data density.

To overcome the difficulty on strong data density regularities assumptions, Chen et al. (2023a) suggest to interleave steps of the discretized probability flow ODEs with Langevin diffusion correctors using the estimated score, and get a better convergence guarantee than SDEs thanks to the $\mathcal{C}^1$ trajectory for ODEs comparing to the $\mathcal{C}^{\frac{1}{2}-}$ trajectory for SDEs. This approach only need to assume the data density to be $L$-smooth.

## 2 PRELIMINARY

### 2.1 DIFFUSION MODELS

Consistency models are heavily relied on the denoising diffusion probabilistic modeling (DDPM). We start with a forward process defined in $\mathbb{R}^d$, which is expressed as a stochastic differential equation

$$d\boldsymbol{x}_t = \boldsymbol{\mu}(\boldsymbol{x}_t, t)\mathrm{d}t + \sigma(t)d\boldsymbol{w}_t \tag{1}$$

where $t \in [0, T], T > 0$ is a fixed constant, $\boldsymbol{\mu}(\cdot, \cdot)$ and $\sigma(\cdot)$ are the drift and diffusion coefficients respectively, and $\{\boldsymbol{w}_t\}_{t \in [0,T]}$ denotes the $d$-dimensional standard Brownian motion. Denote the disbribution of $\boldsymbol{x}_t$ as $p_t(\boldsymbol{x})$, therefore $p_0(\boldsymbol{x}) = p_{\text{data}}(\boldsymbol{x})$. A remarkable property is the existence of an ordinary differential equation dubbed the *probability flow ODE*, whose solution trajectories sampled at $t$ are distributed according to $p_t(\boldsymbol{x})$:

$$\mathrm{d}\boldsymbol{x}_t = \left[\boldsymbol{\mu}(\boldsymbol{x}_t, t) - \frac{1}{2}\sigma(t)^2 \nabla \log p_t(\boldsymbol{x}_t)\right] dt \tag{2}$$

here $\nabla \log p_t(\boldsymbol{x})$ is the *score function* of $p_t(\boldsymbol{x})$.

For clarity, we consider the simplest possible noise schedule choice, which is the Ornstein-Uhlenbeck (OU) process as in Chen et al. (2023a), where $\mu(\boldsymbol{x}, t) = -\boldsymbol{x}$ and $\sigma(t) \equiv \sqrt{2}$,

$$\mathrm{d}\boldsymbol{x}_t = -\boldsymbol{x}_t \mathrm{d}t + \sqrt{2}\mathrm{d}\boldsymbol{w}_t, \boldsymbol{x}_0 \sim p_{\mathrm{data}}, \tag{3}$$

The corresponding backward ODE is

$$\mathrm{d}\boldsymbol{x}_t = (-\boldsymbol{x}_t - \nabla \log p_t(\boldsymbol{x}))\mathrm{d}t. \tag{4}$$

In this case we have

$$p_t(\boldsymbol{x}) = e^{dt}p_{\mathrm{data}}(e^t\boldsymbol{x}) * \mathcal{N}(\boldsymbol{0}, (1 - e^{-2t})\boldsymbol{I}_d), \tag{5}$$

where $*$ denotes the convolution operator. We take $\pi(\boldsymbol{x}) = \mathcal{N}(\boldsymbol{0}, \boldsymbol{I}_d)$, which is a tractable Gaussian distribution close to $p_T(\boldsymbol{x})$. For sampling, we first train a score model $\boldsymbol{s}_\phi \approx \nabla \log p_t(\boldsymbol{x})$ via score matching (Hyvärinen (2005); Song & Ermon (2019); Ho et al. (2020)), then plug into equation 4 to get the empirical estimation of the PF ODE, which takes the form of

$$\mathrm{d}\boldsymbol{x}_t = (-\boldsymbol{x}_t - \boldsymbol{s}_\phi(\boldsymbol{x}_t, t))\mathrm{d}t. \tag{6}$$

We call equation 6 the *empirical PF ODE*. Denote the distribution of $\boldsymbol{x}_t$ in equation 6 as $q_t(\boldsymbol{x})$. Empirical PF ODE gradually transforms $q_T(\boldsymbol{x}) = \pi(\boldsymbol{x})$ into $q_0(\boldsymbol{x})$, which can be viewed as an approximation of $p_{\mathrm{data}}(\boldsymbol{x})$.

## 2.2 CONSISTENCY MODELS

For any ordinary differential equation defined on $\mathbb{R}^d$ with vector field $\boldsymbol{v} : \mathbb{R}^d \times \mathbb{R}^+ \to \mathbb{R}^d$,

$$\mathrm{d}\boldsymbol{x}_t = \boldsymbol{v}(\boldsymbol{x}_t, t)\mathrm{d}t,$$

we may define the associate backward mapping $\boldsymbol{f}^{\boldsymbol{v}} : \mathbb{R}^d \times \mathbb{R}^+ \to \mathbb{R}^d$ such that

$$\boldsymbol{f}^{\boldsymbol{v}}(\boldsymbol{x}_t, t) = \boldsymbol{x}_\delta. \tag{7}$$

with an early-stopping time $\delta > 0$. Under mild conditions on $\boldsymbol{v}$, such a mapping $\boldsymbol{f}^{\boldsymbol{v}}$ exists for any $t \in \mathbb{R}^+$, and is smoothly relied on $\boldsymbol{x}$ and $t$. Note that equation 7 is equivalent to the following conditions

$$\boldsymbol{f}^{\boldsymbol{v}}(\boldsymbol{x}_t, t) = \boldsymbol{f}^{\boldsymbol{v}}(\boldsymbol{x}_s, \tau), \forall 0 \le \tau, t \le T, \text{ and}$$

$$\boldsymbol{f}^{\boldsymbol{v}}(\boldsymbol{x}, \delta) = \boldsymbol{x}, \forall \boldsymbol{x} \in \mathbb{R}^d$$

which playing the essential role in constructing consistency loss.

Now let us take $\boldsymbol{v}^{\mathrm{ex}}(\boldsymbol{x}, t) = -\boldsymbol{x} - \nabla \log p_t(\boldsymbol{x})$ and $\boldsymbol{v}^{\mathrm{em}}(\boldsymbol{x}, t) = -\boldsymbol{x} - \boldsymbol{s}_\phi(\boldsymbol{x}_t, t)$, and denote the corresponding backward mapping function as $\boldsymbol{f}^{\mathrm{ex}}$, for exact vector field $\boldsymbol{v}^{\mathrm{ex}}$, and $\boldsymbol{f}^{\mathrm{em}}$, for empirical vector field $\boldsymbol{v}^{\mathrm{em}}$, respectively. We aim to construct a parametric model $\boldsymbol{f}_\theta$ to approximate $\boldsymbol{f}^{\mathrm{ex}}$. Song et al. (2023) first implement the boundary condition using the skip connection,

$$\boldsymbol{f}_\theta(\boldsymbol{x}, t) = c_{\mathrm{skip}}(t)\boldsymbol{x} + c_{\mathrm{out}}(t)F_\theta(\boldsymbol{x}, t)$$

with differentiable $c_{\mathrm{skip}}(t), c_{\mathrm{out}}(t)$ such that $c_{\mathrm{skip}}(\delta) = 1, c_{\mathrm{out}}(\delta) = 0$, and then define the following Consistency Distillation object:

$$\mathcal{L}_{\mathrm{CD}}^N(\boldsymbol{\theta}, \boldsymbol{\theta}^-; \phi) := \mathbb{E}[\lambda(t_n) \| \boldsymbol{f}_\theta(\boldsymbol{x}_{t_{n+1}}, t_{n+1}) - \boldsymbol{f}_{\boldsymbol{\theta}^-}(\hat{\boldsymbol{x}}_{t_n}^\phi, t_n) \|_2^2], \tag{8}$$

where $0 < t_1 = \delta < t_2 \cdots < t_N = T$, $n$ uniformly distributed over $\{1, 2, \cdots, N-1\}$, $\boldsymbol{x} \sim p_{\mathrm{data}}$, $\boldsymbol{x}_{t_{n+1}} = e^{-t}\boldsymbol{x} + \sqrt{1 - e^{-2t}}\boldsymbol{\xi}, \boldsymbol{\xi} \sim \mathcal{N}(\boldsymbol{0}, \boldsymbol{I}_d)$. Here $\hat{\boldsymbol{x}}_{t_n}^\phi$ is calculated by

$$\hat{\boldsymbol{x}}_{t_n}^\phi := \Phi(\boldsymbol{x}_{t_{n+1}}, t_{n+1}, t_n; \phi) \tag{9}$$

where $\Phi(\cdots; \phi)$ represents the update function of a ODE solver applied to the empirical PF ODE 6. In our noise scheduler 3, We may use the exponential integrator (i.e., exactly integrating the linear part),

$$\hat{\boldsymbol{x}}_{t_n}^\phi = e^{t_{n+1} - t_n}\boldsymbol{x}_{t_{n+1}} + (e^{t_{n+1} - t_n} - 1)\boldsymbol{s}_\phi(\boldsymbol{x}_{t_{n+1}}, t_{n+1}). \tag{10}$$

For simplicity, we assume $\lambda(t_n) \equiv 1$, and only consider the square of $l_2$ distance to build the loss metric, Song et al. (2023) also considered other distance metric such as $l_1$ distance $\|\boldsymbol{x} - \boldsymbol{y}\|_1$, and the Learned Perceptual Image Patch Similarity (LPIPS, Zhang et al. (2018)).

To stabilize the training process, Song et al. (2023) introduce an additional parameter $\boldsymbol{\theta}^-$ and update it by an exponential moving average (EMA) strategy. That is, given a decay rate $0 \le \mu < 1$, the author perform the following update after each optimizaiton step: $\boldsymbol{\theta}^- = \mathrm{stopgrad}(\mu\boldsymbol{\theta}^- + (1-\mu)\boldsymbol{\theta})$.

Besides the distillation strategy that needs an existing score model $\boldsymbol{s}_{\phi}$, Song et al. (2023) also introduced a way to train without any pre-trained score models called the Consistency Training (CT) objective. We refer Lemma 15 for the expression of CT objective under the OU scheduler 5 and the exponential integrator 9.

Song et al. (2023) gave a asymptotic analysis on the approximation error in their original work. If $\mathcal{L}_{\mathrm{CD}}^N(\boldsymbol{\theta}, \boldsymbol{\theta}; \phi) = 0$, we have $\sup_{n,\boldsymbol{x}} \|\boldsymbol{f}_{\boldsymbol{\theta}}(\boldsymbol{x}, t_n) - \boldsymbol{f}^{\mathrm{em}}(\boldsymbol{x}, t_n)\|_2 = O((\Delta t)^p)$ when the numerical integrator has local error uniformly bounded by $O((\Delta t)^{p+1})$ with $p \ge 1$. However, when $\mathcal{L}_{\mathrm{CD}}^N \ne 0$, we also need a quantitative analysis on how far between $\boldsymbol{f}_{\boldsymbol{\theta}}$ and $\boldsymbol{f}^{\mathrm{em}}$, and further, a quantitative analysis between $\boldsymbol{f}_{\boldsymbol{\theta}}$ and $\boldsymbol{f}^{\mathrm{ex}}$, and thus the distance between generated distribution and true data distribution.

## 3 MAIN RESULTS

In this section, we formally state our assumptions and our main results. We denote $\boldsymbol{f}_{\boldsymbol{\theta},t}(\boldsymbol{x}) = \boldsymbol{f}_{\boldsymbol{\theta}}(\boldsymbol{x}, t)$ to emphasize the mapping over $\boldsymbol{x}$ at time $t$. We summarized definition of some notations in the Appendix A.

### 3.1 ASSUMPTIONS

We assume the following mild conditions on the data distribution $p_{\mathrm{data}}$.

**Assumption 1.** *The data distribution has finite $2^{nd}$ moment, that is, $\mathbb{E}_{\boldsymbol{x}_0 \sim p_{\mathrm{data}}}[\|\boldsymbol{x}_0\|_2^2] = \mathfrak{m}^2 < \infty$.*

**Assumption 2.** *The score function $\nabla \log p_t(\boldsymbol{x})$ is Lipschitz on the variable $\boldsymbol{x}$ with Lipschitz constant $L_s \ge 1$, $\forall t \in [0, T]$.*

This two assumptions are standard and has been used in prior works Block et al. (2020); Lee et al. (2022a;b); Chen et al. (2022). As Lee et al. (2022b); Chen et al. (2022), we do not assume Lipschitzness of the score estimate; unlike Block et al. (2020); Bortoli et al. (2021), we do not assume any convexity or dissipativity assumptions on the potential $U = -\log(p_{\mathrm{data}})$, and unlike Lee et al. (2022a) we do not assume $p_{\mathrm{data}}$ satisfies a log-Sobolev inequality. Thus our assumptions are general enough to cover the highly non-log-concave data distributions. Our assumption could be further weaken to only be compactly supported, which will be further discussed in 3.4.

We also assume bounds on the score estimation error and consistency error.

**Assumption 3.** *Assume $\mathbb{E}_{\boldsymbol{x}_{t_n} \sim p_{t_n}}[\|\boldsymbol{s}_{\phi}(\boldsymbol{x}_{t_n}, t_n) - \nabla \log p_{t_n}(\boldsymbol{x}_{t_n})\|_2^2] \le \varepsilon_{sc}^2, \forall n \in [\![1, N]\!]$.*

**Assumption 4.** *Assume $\mathbb{E}_{\boldsymbol{x}_{t_n} \sim p_{t_n}}[\|\boldsymbol{f}_{\boldsymbol{\theta}}(\boldsymbol{x}_{t_{n+1}}, t_{n+1}) - \boldsymbol{f}_{\boldsymbol{\theta}}(\hat{\boldsymbol{x}}_{t_n}^{\phi}, t_n)\|_2^2] \le \varepsilon_{cm}^2(t_{n+1} - t_n)^2, \forall n \in [\![1, N-1]\!]$, where $\hat{\boldsymbol{x}}_{t_n}^{\phi}$ is the exponential integrator defined as in equation 10.*

The score estimation error is the same as in Lee et al. (2022a); Chen et al. (2022). As discussed in Section 2, these two assumption are nature and realistic in light of the derivation of the score matching objective and consistency distillation object.

The following Lipschitz condition for the consistency model is nature and has been used in prior work ( Song et al. (2023), Theorem 1).

**Assumption 5.** *The consistency model $\boldsymbol{f}_{\boldsymbol{\theta}}(\boldsymbol{x}, t_n)$ is Lipschitz on the variable $\boldsymbol{x}$ with Lipschitz constant $L_f > 1$, $\forall n \in [\![1, N]\!]$.*

For technique reason, we divide our discretization schedule into two stages: in the first stage, which lasts from $T$ to $h$, we keep the step size equal to $h$; in the second stage, which lasts from $h$ to $\delta$, we take a geometric reducing sequence $2^{-1}h, 2^{-2}h, \cdots$ until $2^{-l}h \le \delta$ for some $l \ge 1$.

**Assumption 6.** *Assume the discretization schedule* $0 < \delta = t_1 < t_2 < \cdots < t_N = T$, $h_k = t_{k+1} - t_k$ *to equation 6 is divided into two stages:*

1. $h_k \equiv h$ *for all* $k \in [\![N_1, N-1]\!]$, *and* $(N - N_1 - 1)h < T \le (N - N_1)h$;

2. $h_k = 2^{-(N_1-k)}h = \frac{h_{k+1}}{2}$ *for* $k \in [\![1, N_1 - 1]\!]$, $N_1$ *satisfies* $h_2 = 2^{-(N_1-2)}h \le 2\delta$.

*note that in this case* $h_1 = T - \sum_{k=2}^{N} h_k - \delta \le h - (1 - 2^{-(N_1-2)})h - \delta \le \delta$, *and* $t_{N_1} \le h$

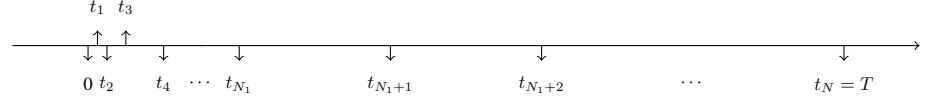

Figure 1: Illustration of the discretization schedule in Assumption 6

## 3.2  $W_2$ ERROR GUARANTEE FOR ONE-STEP CONSISTENCY GENERATING

In this section we introduce our main results. The first result bounds the Consistency Model estimation error in an expectation mean.

**Theorem 2** (see Section B.1 of Appendix)**.** *Under Assumptions 1-5, assume we choose* $\Phi$ *as the exponential integrator, and assume the timestep schedule satisfies assumption 6, for* $1 \le n \le N-1$:

$$\left(\mathbb{E}_{\boldsymbol{x}_{t_n} \sim p_{t_n}}[\|\boldsymbol{f}_{\boldsymbol{\theta}}(\boldsymbol{x}_{t_n}, t_n) - \boldsymbol{f}^{ex}(\boldsymbol{x}_{t_n}, t_n)\|_2^2]\right)^{1/2} \lesssim t_n(\varepsilon_{cm} + L_f \varepsilon_{sc} + L_f L_s^{\frac{3}{2}} d^{\frac{1}{2}} h) + t_n^{\frac{1}{2}} L_f L_s d^{\frac{1}{2}} h.$$

Now we can get our first theorem that analysis the $W_2$ distance after Consistency Model mapping.

**Theorem 3** (see Section B.2 of Appendix)**.** *Under Assumptions 1-6, let* $\mu(\boldsymbol{x})$ *be any probability density,* $p_t(\boldsymbol{x}) = e^{dt}p_{\text{data}}(e^t\boldsymbol{x}) * \mathcal{N}(\boldsymbol{0}, (1 - e^{-2t})\boldsymbol{I}_d)$, *then the following estimation holds,*

$$W_2(\boldsymbol{f}_{\boldsymbol{\theta}, t_n} \sharp \mu, p_\delta) \lesssim L_f W_2(\mu, p_{t_n}) + t_n(\varepsilon_{cm} + L_f \varepsilon_{sc} + L_f L_s^{\frac{3}{2}} d^{\frac{1}{2}} h) + t_n^{\frac{1}{2}} L_f L_s d^{\frac{1}{2}} h. \tag{11}$$

As a consequence, we directly get the one-step generation error.

**Corollary 4** (see Section B.2 of Appendix)**.** *Under Assumptions 1-6, when* $T > L_s^{-1}$, *the one-step generating error is bounded as follows,*

$$W_2(\boldsymbol{f}_{\boldsymbol{\theta}, T} \sharp \mathcal{N}(\boldsymbol{0}, \boldsymbol{I}_d), p_{\text{data}}) \lesssim (d^{\frac{1}{2}} \vee \mathfrak{m})L_f e^{-T} + T(\varepsilon_{cm} + L_f \varepsilon_{sc} + L_f L_s^{\frac{3}{2}} d^{\frac{1}{2}} h) + (d^{\frac{1}{2}} \vee \mathfrak{m})\delta^{\frac{1}{2}}. \tag{12}$$

*In particular, for any* $\varepsilon > 0$, *if we set* $\delta \asymp \frac{\varepsilon^2}{d \vee \mathfrak{m}^2}$, $T \ge O(\log(\frac{L_f(\sqrt{d} \vee R)}{\varepsilon}))$, *step size* $h = O(\frac{\varepsilon}{T L_f L_s^{3/2} d^{1/2}})$, $\varepsilon_{cm} = O(\frac{\varepsilon}{T})$, $\varepsilon_{sc} = O(\frac{\varepsilon}{L_f T})$, *we can guarantee* $W_2(\boldsymbol{f}_{\boldsymbol{\theta}, T} \sharp \mathcal{N}(\boldsymbol{0}, \boldsymbol{I}_d), p_{\text{data}}) \lesssim \varepsilon$.

We remark that our discretization complexity $N = O(\frac{T}{h}\log(\frac{1}{h})) = O(\frac{L_f L_s^{3/2} d^{1/2}}{\varepsilon})$ matches state-of-the-art complexity for ODE-type SGMs Chen et al. (2023a;b). This provides some evidence that our descretization bounds are of the correct order.

Note that the error bound in 12 relied on the final time $T$, consistency error $\varepsilon_{\text{cm}}$, score error $\varepsilon_{\text{sc}}$ and step size $h$. Actually we can refine the error and reduce the linear dependency on $T$ to log dependency by Multistep Consistency Sampling that will be introduced in the next section.

## 3.3  MULTISTEP CONSISTENCY SAMPLING CAN REDUCE THE $W_2$ ERROR

Now let us analysis the effect of Multistep Consistency Sampling introduced in original CM work, Algorithm 1, Song et al. (2023), which has been introduced to improve the sample quality by alternating denoising and noise injection steps. Given a sequence of time points $T = t_{n_1} \ge t_{n_2} \ge \cdots$, adapted to the OU noise scheduler 3 , the generating procedure can be written as

$$\boldsymbol{z}_1 := \boldsymbol{f}_{\boldsymbol{\theta}}(\boldsymbol{\xi}_1, T),$$
$$\boldsymbol{u}_k := e^{-(t_{n_k} - \delta)}\boldsymbol{z}_{k-1} + \sqrt{1 - e^{-2(t_{n_k} - \delta)}}\boldsymbol{\xi}_k,$$
$$\boldsymbol{z}_k := \boldsymbol{f}_{\boldsymbol{\theta}}(\boldsymbol{u}_k, t_{n_k}), \tag{13}$$

with $\boldsymbol{\xi}_i$ i.i.d. $N(\boldsymbol{0}, \boldsymbol{I}_d)$ distributed, and $q_k := \text{law}(\boldsymbol{z}_k)$ satisfies the following relationship:

$$\begin{aligned}
q_1 &= \boldsymbol{f}_{\boldsymbol{\theta},T} \sharp \mathcal{N}(\boldsymbol{0}, \boldsymbol{I}_d), \\
\mu_k &:= \left( e^{d(t_{n_k}-\delta)} q_{k-1}(e^{(t_{n_k}-\delta)} \boldsymbol{x}) \right) * \mathcal{N}(\boldsymbol{0}, (1 - e^{-2(t_{n_k}-\delta)}) \boldsymbol{I}_d). \\
q_k &= \boldsymbol{f}_{\boldsymbol{\theta},t_{n_k}} \sharp \mu_k,
\end{aligned} \tag{14}$$

We thus have the following upper bound of the $W_2$ distance between $q_k$ and $p_\delta$.

**Corollary 5** (see Section B.3 of Appendix). *Under Assumptions 1-6, when $T > L_s^{-1}$, the $W_2$ distance between $q_k$ and $p_\delta$ can be controlled by $q_{k-1}$ and $p_\delta$ as follows,*

$$W_2(q_k, p_\delta) \lesssim L_f e^{-t_{n_k}} W_2(q_{k-1}, p_\delta) + t_{n_k} (\varepsilon_{cm} + L_f \varepsilon_{sc} + L_f L_s^{\frac{3}{2}} d^{\frac{1}{2}} h). \tag{15}$$

Note that in equation 15, we have an exponentially small multiplier $e^{-t_{n_k}}$ gradually reduce the error introduced from the previous steps, and a $t_{n_k}$-linear term representing the error introduced from the current step. By choosing a suitable time schedule $\{t_{n_k}\}_{k \geq 1}$, we can get a finer bound in $W_2$.

**Corollary 6** (see Section B.3 of Appendix). *Under Assumptions 1-6, there exists $T \geq t_{\hat{n}} \geq \max(\log(2L_f) + \delta, L_s^{-1})$, $\hat{n} \in [\![1, N]\!]$, such that when taking $n_k \equiv \hat{n}$ for all $k$,*

$$W_2(q_k, p_{\text{data}}) \lesssim (\log(L_f) + 2^{-k}T)(\varepsilon_{cm} + L_f \varepsilon_{sc} + L_f L_s^{\frac{3}{2}} d^{\frac{1}{2}} h) + 2^{-k}(d^{\frac{1}{2}} \vee \mathfrak{m}) L_f e^{-T} + (d^{\frac{1}{2}} \vee \mathfrak{m}) \delta^{\frac{1}{2}}.$$

*Thus, for any $\varepsilon > 0$, if we set $\delta \asymp \frac{\varepsilon^2}{d \vee \mathfrak{m}^2}$, $k = O(\log(T \vee (\frac{(d \vee \mathfrak{m}^2) L_f}{\varepsilon})))$, $h = O(\frac{\varepsilon}{\log(L_f) L_f L_s^{3/2} d^{1/2}})$, $\varepsilon_{cm} = O(\frac{\varepsilon}{\log(L_f)})$, $\varepsilon_{sc} = O(\frac{\varepsilon}{L_f \log(L_f)})$, we can guarantee $W_2(q_k, p_{\text{data}}) \lesssim \varepsilon$.*

**Remark 1.** *Comparing the result between multistep sampling error 6 and one step sampling error 4, the main improvement of multistep sampling is getting rid of the linear dependency from $T$. In one step sampling, one should take the step size smaller, and train the consistency model and score model better. Besides, multistep sampling 6 only requires $T \geq \max(\log(2L_f) + \delta, L_s^{-1})$, while one step sampling 4 requires $T \geq O(\log(\frac{L_f(\sqrt{d} \vee R)}{\varepsilon}))$, which is an added benefit that multistep sampling requires lower training complexity comparing to one step sampling.*

## 3.4 $W_2$ CONVERGENCE GUARANTEE FOR ARBITRARILY DATA DISTRIBUTIONS WITH BOUNDED SUPPORT

In this section, we consider a much more general case: in fact for any compactly supported distribution $p_{\text{data}}$, $\text{supp } p_{\text{data}} \subseteq B(\boldsymbol{0}, R)$, for any $t_0$, we can get a positive $L_s(t_0)$, such that Assumption 2 is satisfied for any $t > t_0$ with Lipschitz constant $L_s(t_0)$. This include a wide range of situations even when $p$ do not have smooth density w.r.t. Lebesgue measure such as when $p$ supported on a lower-dimensional submanifold of $\mathbb{R}^d$, which recently investigated in Bortoli (2022); Lee et al. (2022b); Chen et al. (2022).

Namely, based on the following lemma, we can conduct regularity properties for the score functions.

**Lemma 7** (see Section B.4 of Appendix). *Suppose that $\text{supp } p_{\text{data}} \subseteq B(\boldsymbol{0}, R)$ where $R \geq 1$, and let $p_t$ denote the law of the OU process at time $t$, started at $p$: that is, $p_t(\boldsymbol{x}) = e^{dt} p_{\text{data}}(e^t \boldsymbol{x}) * \mathcal{N}(\boldsymbol{0}, (1 - e^{-2t}) \boldsymbol{I}_d)$. Then the Hessian of the score function satisfies:*

$$\|\nabla^2 \log p_t(\boldsymbol{x})\|_{op} \leq \frac{e^{-2t} R^2}{(1 - e^{-2t})^2} + \frac{1}{1 - e^{-2t}}.$$

Note that in our proof of Corollary 4 and 6, we only use the Assumption 2 over $t \in [\delta, T]$. Combining Lemma 7, we immediately get the following corollary.

**Corollary 8** (see Section B.4 of Appendix). *Under Assumptions 3-6, suppose that $\text{supp } p_{\text{data}} \subseteq B(\boldsymbol{0}, R)$ where $R \geq 1$. Let $\delta \asymp \frac{\varepsilon^2}{R^2 \vee d}$, then (1) the one-step generating error satisfies $W_2(q_1, p_{\text{data}}) \lesssim \varepsilon$, provided that $T = O(\log(\frac{L_f(\sqrt{d} \vee R)}{\varepsilon}))$, $h = O(\frac{\varepsilon^7}{d^{1/2} R^3 (R^6 \vee d^3) L_f T})$, $\varepsilon_{cm} = O(\frac{\varepsilon}{T})$, $\varepsilon_{sc} = O(\frac{\varepsilon}{L_f T})$; (2) the multi-step generating error satisfies $W_2(q_k, p_{\text{data}}) \lesssim \varepsilon$, provided that $k = O(\log(T \vee (\frac{(d \vee \mathfrak{m}^2) L_f L_s}{\varepsilon})))$, $T = O(\max(\log(2L_f) + \delta, L_s^{-1}))$, $h = O(\frac{\varepsilon^7}{d^{1/2} R^3 (R^6 \vee d^3) L_f \log(L_f)})$, $\varepsilon_{cm} = O(\frac{\varepsilon}{\log(L_f)})$, $\varepsilon_{sc} = O(\frac{\varepsilon}{L_f \log(L_f)})$.*

## 3.5 BOUNDING THE TV ERROR

In the sections before, we have showed that the generated distribution of Consistency Models are close to the true data distribution in the metric of Wasserstein-2 distance. When we turn to the Total Variational (TV) distance, however, the error bound is deficient as the situation for the probability flow ODEs, in contrast to the situation for the probability flow SDEs. Here we introduce two operations that can further bound the TV error.

### 3.5.1 BOUNDING THE TV ERROR BY FORWARD OU PROCESS

Let the forward OU process be

$$\mathrm{d}\boldsymbol{x}_t = -\boldsymbol{x}_t\mathrm{d}t + \sqrt{2}\mathrm{d}\boldsymbol{w}_t, \tag{16}$$

and denote the associate Markov kernel as $P_{\mathrm{OU}}^s$, that is, if $\boldsymbol{x}_t \sim p$, $\boldsymbol{x}_{t+s} \sim pP_{\mathrm{OU}}^s$. Let $q$ be the output of our Consistency Models, either the one step consistency sampling result, or the k-th multistep consistency sampling result. To control the TV error, we smooth the generated sample by the forward OU process with a small time that is the same as the early stopping time $\delta$, and then we can get the TV distence between $qP_{\mathrm{OU}}^\delta$ and $p_{\mathrm{data}}$:

**Corollary 9** (see Section B.5 of Appendix). *Under Assumptions 1-6, suppose $q$ is: (1) the one step consistency sampling result, $q = q_1 = \boldsymbol{f}_{\boldsymbol{\theta},T}\sharp\mathcal{N}(\boldsymbol{0}, \boldsymbol{I}_d)$; (2) the k-th multistep consistency sampling result, $q = q_k$ defined as in equation 14 with multistep schedule as in Corollary 6. Choose the early stopping time $\delta \asymp \frac{\varepsilon^2}{L_s^2(d\vee\mathfrak{m}^2)}$ for some $\varepsilon > 0$, then if $T \geq \max(\log(2L_f) + \delta, L_s^{-1})$,*

$$TV(q_1P_{OU}^\delta, p_{\mathrm{data}})$$
$$\lesssim \frac{L_sL_f(d \vee \mathfrak{m}^2)}{\varepsilon}e^{-T} + \frac{L_s(d^{\frac{1}{2}} \vee \mathfrak{m})}{\varepsilon}T(\varepsilon_{cm} + L_f\varepsilon_{sc} + L_fL_s^{\frac{3}{2}}d^{\frac{1}{2}}h) + \varepsilon,$$
$$TV(q_kP_{OU}^\delta, p_{\mathrm{data}})$$
$$\lesssim \frac{L_s(d^{\frac{1}{2}} \vee \mathfrak{m})}{\varepsilon}\left((\log L_f + \frac{T}{2^k})(\varepsilon_{cm} + L_f\varepsilon_{sc} + L_fL_s^{\frac{3}{2}}d^{\frac{1}{2}}h) + \frac{(d^{\frac{1}{2}} \vee \mathfrak{m})L_f}{2^ke^T}\right) + \varepsilon.$$

*In particular,*

1. *If we set $T = O(\log(\frac{(d\vee\mathfrak{m}^2)L_fL_s}{\varepsilon^4}))$, $h = O(\frac{\varepsilon^2}{L_fL_s^{5/2}(d\vee\mathfrak{m}^2)T})$, and if $\varepsilon_{sc} \leq O(\frac{\varepsilon^2}{TL_fL_s(d^{1/2}\vee\mathfrak{m})})$, $\varepsilon_{cm} \leq O(\frac{\varepsilon^2}{TL_s(d^{1/2}\vee\mathfrak{m})})$, then we can guarantee TV error $O(\varepsilon)$ with one step prediction and one additional OU correction (with no NN evaluation);*

2. *If we set $k = O(\log(T \vee (\frac{(d\vee\mathfrak{m}^2)L_fL_s}{\varepsilon})))$, $h = O(\frac{\varepsilon^2}{\log(L_f)L_fL_s^{5/2}(d\vee\mathfrak{m}^2)})$ and if $\varepsilon_{sc} \leq O(\frac{\varepsilon^2}{\log(L_f)L_fL_s(d^{1/2}\vee\mathfrak{m})})$, $\varepsilon_{cm} \leq O(\frac{\varepsilon^2}{\log(L_f)L_s(d^{1/2}\vee\mathfrak{m})})$, then we can guarantee TV error $O(\varepsilon)$ with k steps prediction and one additional OU correction (with no NN evaluation).*

### 3.5.2 BOUNDING THE TV ERROR BY UNDERDAMPED LANGEVIN CORRECTOR

We may adopt the idea from Chen et al. (2023a), who introduce the Langevin-correcting procedure into the probability flow ODEs to get a TV error guarantee.

The Langevin dynamics for correcting purpose is defined as follows: let $p$ be a distribution over $\mathbb{R}^d$, and write $U$ as a shorthand for the potential $-\log p$.

Given a friction parameter $\gamma > 0$, consider the following discretized process with step size $\tau$, where $-\nabla U$ is replaced by a score estimator $\boldsymbol{s}$. Let $(\hat{z}_t, \hat{v}_t)_{t\geq 0}$ over $\mathbb{R}^d \otimes \mathbb{R}^d$ be given by

$$\mathrm{d}\hat{\boldsymbol{z}}_t = \hat{\boldsymbol{v}}_t\mathrm{d}t,$$
$$\mathrm{d}\hat{\boldsymbol{v}}_t = (\boldsymbol{s}(\hat{\boldsymbol{z}}_{\lfloor t/\tau\rfloor\tau}) - \gamma\hat{\boldsymbol{v}}_t)\mathrm{d}t + \sqrt{2\gamma}\mathrm{d}\boldsymbol{w}_t. \tag{17}$$

Denote the Markov kernel $\hat{P}_{\mathrm{ULMC}}$ to be defined by the equation 17, that is, if $(\hat{z}_{k\tau}, \hat{v}_{k\tau}) \sim \boldsymbol{\mu}$ for some $\boldsymbol{\mu}$ be a distribution over $\mathbb{R}^{d\times d}$, $(\hat{z}_{(k+1)\tau}, \hat{v}_{(k+1)\tau}) \sim \boldsymbol{\mu}\hat{P}_{\mathrm{ULMC}}$. We denote the k-th composition

$\hat{P}_{\text{ULMC}}^k = \hat{P}_{\text{ULMC}} \circ \hat{P}_{\text{ULMC}}^{k-1}$, and $\boldsymbol{q} = q \otimes \mathcal{N}(0, \boldsymbol{I}_d), \boldsymbol{p} = p \otimes \mathcal{N}(0, \boldsymbol{I}_d)$. In what follows, we abuse the notation as follows. Given a distribution $q$ on $\mathbb{R}^d$, we write $q\hat{P}_{\text{ULMC}}$ to denote the projection onto the $\boldsymbol{z}-$coordinates of $\boldsymbol{q}\hat{P}_{\text{ULMC}}$. It's obvious that

$$\text{TV}(q\hat{P}_{\text{ULMC}}^N, p) \leq \text{TV}(\boldsymbol{q}\hat{P}_{\text{ULMC}}^N, \boldsymbol{p}).$$

Now let's take $p = p_{\text{data}}$ as the data distribution, and $q = q_k$ for some $k \geq 1$ as the output distribution of $k$-th multistep consistency sampling defined as in equation 14. We can use the score model $\boldsymbol{s}_\phi(\boldsymbol{x}, \delta) \approx \nabla \log p_\delta(\boldsymbol{x})$ to do corrector steps. The end-to-end error now can be written as follows:

**Corollary 10** (see Section B.6 of Appendix). *Under Assumptions 1-6, suppose q is: (1) the one step consistency sampling result, $q = q_1 = \boldsymbol{f}_{\boldsymbol{\theta},T}\sharp\mathcal{N}(\boldsymbol{0}, \boldsymbol{I}_d)$; (2) the k-th multistep consistency sampling result, $q = q_k$ defined as in equation 14 with multistep schedule as in Corollary 6. Choose $\gamma \asymp L_s$, and $\delta \asymp \frac{\varepsilon^2}{L_s^2(d\vee\mathfrak{m}^2)}$ for some $\varepsilon > 0$, then if $T \geq \max(\log(2L_f) + \delta, L_s^{-1}), N\tau \asymp \frac{1}{\sqrt{L_s}}$,*

$TV(q_1\hat{P}_{ULMC}^N, p_{\text{data}})$

$$\lesssim (d^{\frac{1}{2}} \vee \mathfrak{m})L_f L_s^{\frac{1}{2}} e^{-T} + TL_s^{\frac{1}{2}}(\varepsilon_{cm} + L_f\varepsilon_{sc} + L_f L_s^{\frac{3}{2}} d^{\frac{1}{2}} h) + L_s^{-\frac{1}{2}}\varepsilon_{sc} + L_s^{\frac{1}{2}} d^{\frac{1}{2}} \tau + \varepsilon,$$

$TV(q_k\hat{P}_{ULMC}^N, p_{\text{data}})$

$$\lesssim (\log(L_f) + \frac{T}{2^k})L_s^{\frac{1}{2}}(\varepsilon_{cm} + L_f\varepsilon_{sc} + L_f L_s^{\frac{3}{2}} d^{\frac{1}{2}} h) + \frac{(d^{\frac{1}{2}} \vee \mathfrak{m})L_s^{\frac{1}{2}} L_f}{2^k e^T} + L_s^{-\frac{1}{2}}\varepsilon_{sc} + L_s^{\frac{1}{2}} d^{\frac{1}{2}} \tau + \varepsilon.$$

*In particular,*

1. *if we set $T = O(\log(\frac{(d\vee\mathfrak{m}^2)L_f^2 L_s}{\varepsilon^2}))$, $h = O(\frac{\varepsilon}{L_f L_s^2 d^{1/2} T})$, $\tau = O(\frac{\varepsilon}{L_s^{1/2} d^{1/2}})$, and if $\varepsilon_{sc} \leq O(\frac{\varepsilon}{TL_f L_s^{1/2}})$, $\varepsilon_{cm} \leq O(\frac{\varepsilon}{TL_s^{1/2}})$, then we can obtain TV error $O(\varepsilon)$ with one step consistency prediction and $O(\frac{\sqrt{d}}{\varepsilon})$ steps correcting;*

2. *if we set $k = O(\log(T \vee (\frac{(d\vee\mathfrak{m}^2)L_f L_s}{\varepsilon})))$, $h = O(\frac{\varepsilon}{\log(L_f)L_f L_s^2 d^{1/2}})$, $\tau = O(\frac{\varepsilon}{L_s^{1/2} d^{1/2}})$ and if $\varepsilon_{sc} \leq O(\frac{\varepsilon}{\log(L_f)L_f L_s^{1/2}})$, $\varepsilon_{cm} \leq O(\frac{\varepsilon}{\log(L_f)L_s^{1/2}})$, then we can obtain TV error $O(\varepsilon)$ with $k$ steps consistency prediction and $O(\frac{\sqrt{d}}{\varepsilon})$ steps correcting.*

**Remark 2.** *In case of missing the score model $\boldsymbol{s}_\phi$ but only remain the consistency model $\boldsymbol{f}_{\boldsymbol{\theta}}$, such as training the consistency model by CT objective 46 without a pretrained score model, we can also recover a score model $\hat{\boldsymbol{s}}_{\boldsymbol{\theta}}$ that approx $\nabla \log p_{t_2}(\boldsymbol{x})$: in fact $\hat{\boldsymbol{s}}_{\boldsymbol{\theta}}(\boldsymbol{x}) := \frac{\boldsymbol{f}_{\boldsymbol{\theta}}(\boldsymbol{x},t_2)-e^{h_1}\boldsymbol{x}}{e^{h_1}-1}$ is a score model that approximate $\nabla \log p_{t_2}(\boldsymbol{x})$ with $L^2$ error $\varepsilon_{cm} + \varepsilon_{sc}$ ( proved in Appendix, Lemma 14). Hence, we may run similar procedure as in Theorem 10, where the output distribution $q_k$ of consistency model should firstly be transformed with forward OU process equation 3 under a small time $h_1 = t_2 - \delta < \delta$, then apply the Underdamped Langevin Corrector Operator with $\hat{\boldsymbol{s}}_{\boldsymbol{\theta}}(\boldsymbol{x})$.*

## 4 CONCLUSIONS AND LIMITATIONS

In this work, we provided a first convergence guarantee for CMs which holds true under realistic assumptions ($L^2$-accurate score and consistency function surrogates; arbitrarily data distributions with smooth densities respect to Lebesgue measure, or bounded distributions) and which scale at most polynomially in all relavent parameters. Our results take a step towards explaining the success of CMs. We also provide theoretical evidence that multistep CM sampling technique can further reduce the error comparing to one step CM sampling .

There are mainly three shortcomings in our work. Firstly, our proofs relied on the lipschitz conditions of the surrogate model $\boldsymbol{f}_{\boldsymbol{\theta}}$, which is somehow unrealistic. We will keep in improving our results by replacing this condition on the exact consistency function $\boldsymbol{f}^{\text{ex}}$ in our future works. Secondly, to bound the TV error, we introduced an additional smoothing procedure after the original CM sampling steps. It would be better to remove this unnatural procedure by introduce new assumptions or techniques. Lastly, we did not address the question of when the score and consistency function can be learned well enough. We believe that the resolution of these problem would shed considerable light on Consistency Models.

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

# A COMMON SYMBOLS

- $*$: the convolution operator defined for two functions $f, g \in L^2(\mathbb{R}^d)$.
  $f * g(\boldsymbol{x}) := \int_{\mathbb{R}^d} f(\boldsymbol{x} - \boldsymbol{u})g(\boldsymbol{u})d\boldsymbol{u}$.

- $\lesssim$: less or similar to. If $a \lesssim b$, it means $a \leq Cb$ for some constant $C$.

- $\sharp$ the push-forward operator associated with a measurable map $f : \mathcal{M} \to \mathcal{N}$. For any measure $\mu$ over $\mathcal{M}$, we may define the push-forward measure $f\sharp\mu$ over $\mathcal{N}$ by: $f\sharp\mu(A) = \mu(f^{-1}(A))$, for any $A$ be measurable set in $\mathcal{N}$.

- $\vee$: take the larger one. $a \vee b = \max(a, b)$;
  $\wedge$: take the smaller one. $a \wedge b = \min(a, b)$.

- $\asymp$: asymptotic to. If $a_n \asymp b_n$, it means $\lim_{n \to \infty} a_n/b_n = C$ for some constant $C$.

- $[\![a, b]\!] := [a, b] \cap \mathbb{Z}$. For example, $[\![1, N]\!] = \{1, 2, 3, \cdots, N\}$, for any $N \in \mathbb{Z}^+$

# B PROOFS

## B.1 PROOFS FOR THEOREM 2

Before we proof our main theorem, we introduce a score perturbation lemma which comes from Lemma 1 in Chen et al. (2023a).

**Lemma 11** (Lemma 1 in Chen et al. (2023a)). *(Score perturbation for $\boldsymbol{x}_t$). Suppose $p_t(\boldsymbol{x}) = e^{dt}p_{\text{data}}(e^t\boldsymbol{x}) * \mathcal{N}(\boldsymbol{0}, (1 - e^{-2t})\boldsymbol{I}_d)$ started at $p_0$, and $\boldsymbol{x}_0 \sim p_0$, $d\boldsymbol{x}_t = -\boldsymbol{x}_t - \nabla \log p_t(\boldsymbol{x})$. Suppose that $\|\nabla^2 \log p_t(\boldsymbol{x})\|_{op} \leq L$ for all $\boldsymbol{x} \in \mathbb{R}^d$ and $t \in [0, T]$, where $L \geq 1$. Then,*

$$\mathbb{E}\left[\left\|\frac{\partial}{\partial_t}\nabla \log p_t(\boldsymbol{x}_t)\right\|_2^2\right] \lesssim L^2 d(L + \frac{1}{t})$$

**Proof of Theorem 2.** We divide the left-hand side by Cauchy-Schwarz inequality as follows: let $\boldsymbol{x}_t$ be the solution to the probability flow ODE 4. Notice that $\boldsymbol{f}^{\text{ex}}(\boldsymbol{x}_{t_n}, t_n) = \boldsymbol{x}_\delta = \boldsymbol{f}_{\boldsymbol{\theta}}(\boldsymbol{x}_{t_1}, t_1)$

$$\left(\mathbb{E}_{\boldsymbol{x}_{t_n} \sim p_{t_n}}[\|\boldsymbol{f}_{\boldsymbol{\theta}}(\boldsymbol{x}_{t_n}, t_n) - \boldsymbol{f}^{\text{ex}}(\boldsymbol{x}_{t_n}, t_n)\|_2^2]\right)^{1/2}$$

$$= \left(\mathbb{E}_{\boldsymbol{x}_{t_n} \sim p_{t_n}}\left[\left\|\sum_{k=1}^{n-1}(\boldsymbol{f}_{\boldsymbol{\theta}}(\boldsymbol{x}_{t_{k+1}}, t_{k+1}) - \boldsymbol{f}_{\boldsymbol{\theta}}(\boldsymbol{x}_{t_k}, t_k))\right\|_2^2\right]\right)^{1/2}$$

$$\leq \sum_{k=1}^{n-1}\left(\mathbb{E}_{\boldsymbol{x}_{t_n} \sim p_{t_n}}\left[\|\boldsymbol{f}_{\boldsymbol{\theta}}(\boldsymbol{x}_{t_{k+1}}, t_{k+1}) - \boldsymbol{f}_{\boldsymbol{\theta}}(\boldsymbol{x}_{t_k}, t_k)\|_2^2\right]\right)^{1/2}$$

$$= \sum_{k=1}^{n-1}\left(\mathbb{E}_{\boldsymbol{x}_{t_n} \sim p_{t_n}}\left[\left\|\boldsymbol{f}_{\boldsymbol{\theta}}(\boldsymbol{x}_{t_{k+1}}, t_{k+1}) - \boldsymbol{f}_{\boldsymbol{\theta}}(\hat{\boldsymbol{x}}_{t_k}^{\phi}, t_k) + \boldsymbol{f}_{\boldsymbol{\theta}}(\hat{\boldsymbol{x}}_{t_k}^{\phi}, t_k) - \boldsymbol{f}_{\boldsymbol{\theta}}(\boldsymbol{x}_{t_k}, t_k)\right\|_2^2\right]\right)^{1/2}$$

$$\leq \sum_{k=1}^{n-1}\left(\mathbb{E}_{\boldsymbol{x}_{t_n} \sim p_{t_n}}\left[\left\|\boldsymbol{f}_{\boldsymbol{\theta}}(\boldsymbol{x}_{t_{k+1}}, t_{k+1}) - \boldsymbol{f}_{\boldsymbol{\theta}}(\hat{\boldsymbol{x}}_{t_k}^{\phi}, t_k)\right\|_2^2\right]\right)^{1/2}$$

$$+ \sum_{k=1}^{n-1}\left(\mathbb{E}_{\boldsymbol{x}_{t_n} \sim p_{t_n}}\left[\left\|\boldsymbol{f}_{\boldsymbol{\theta}}(\hat{\boldsymbol{x}}_{t_k}^{\phi}, t_k) - \boldsymbol{f}_{\boldsymbol{\theta}}(\boldsymbol{x}_{t_k}, t_k)\right\|_2^2\right]\right)^{1/2}$$

$$:= E_1 + E_2 \tag{18}$$

We can bound $E_1$ by assumption 4: note that

$$
\begin{aligned}
E_1 &= \sum_{k=1}^{n-1} \left( \mathbb{E}_{\boldsymbol{x}_{t_n} \sim p_{t_n}} \left[ \left\| \boldsymbol{f_\theta}(\boldsymbol{x}_{t_{k+1}}, t_{k+1}) - \boldsymbol{f_\theta}(\hat{\boldsymbol{x}}_{t_k}^\phi, t_k) \right\|_2^2 \right] \right)^{1/2} \\
&= \sum_{k=1}^{n-1} \left( \mathbb{E}_{\boldsymbol{x}_{t_{k+1}} \sim p_{t_{k+1}}} \left[ \left\| \boldsymbol{f_\theta}(\boldsymbol{x}_{t_{k+1}}, t_{k+1}) - \boldsymbol{f_\theta}(\hat{\boldsymbol{x}}_{t_k}^\phi, t_k) \right\|_2^2 \right] \right)^{1/2} \\
&\leq \varepsilon_{\mathrm{cm}} \sum_{k=1}^{n-1} h_k = \varepsilon_{\mathrm{cm}}(t_n - t_1),
\end{aligned}
\tag{19}
$$

where the last equality we use the fact that when $\boldsymbol{x}_t$ satisfies equation 4 and $\boldsymbol{x}_{t_n} \sim p_{t_n}, \boldsymbol{x}_{t_k} \sim p_{t_k}$ for all $k \leq N$.

Now we turn to bounding the second term. We notice that by Lipschitz assumption 5,

$$
\begin{aligned}
E_2 &= \left( \sum_{k=1}^{n-1} \mathbb{E}_{\boldsymbol{x}_{t_n} \sim p_{t_n}} \left[ \left\| \boldsymbol{f_\theta}(\hat{\boldsymbol{x}}_{t_k}^\phi, t_k) - \boldsymbol{f_\theta}(\boldsymbol{x}_{t_k}, t_k) \right\|_2^2 \right] \right)^{1/2} \\
&\leq \sum_{k=1}^{n-1} L_f \left( \mathbb{E}_{\boldsymbol{x}_{t_n} \sim p_{t_n}} \left[ \| \hat{\boldsymbol{x}}_{t_k}^\phi - \boldsymbol{x}_{t_k} \|_2^2 \right] \right)^{1/2} \\
&= \sum_{k=1}^{n-1} L_f \left( \mathbb{E}_{\boldsymbol{x}_{t_{k+1}} \sim p_{t_{k+1}}} \left[ \| \hat{\boldsymbol{x}}_{t_k}^\phi - \boldsymbol{x}_{t_k} \|_2^2 \right] \right)^{1/2}
\end{aligned}
\tag{20}
$$

Now let us bound the term $\| \hat{\boldsymbol{x}}_{t_k}^\phi - \boldsymbol{x}_{t_k} \|_2^2$. Note that $\hat{\boldsymbol{x}}_{t_k}^\phi$ is the exponential integrator solution to the ODE 6, we have

$$
\begin{aligned}
\mathrm{d}\boldsymbol{x}_t &= -(\boldsymbol{x}_t + \nabla \log p_t(\boldsymbol{x}_t))\mathrm{d}t, \\
\mathrm{d}\hat{\boldsymbol{x}}_t^\phi &= -(\hat{\boldsymbol{x}}_t^\phi + \boldsymbol{s}(\hat{\boldsymbol{x}}_{t_{k+1}}^\phi, t_{k+1}))\mathrm{d}t.
\end{aligned}
\tag{21}
$$

for $t_k \leq t \leq t_{k+1}$ with $\hat{\boldsymbol{x}}_{t_{k+1}}^\phi = \boldsymbol{x}_{t_{k+1}}$. Denote $h_k = t_{k+1} - t_k$, then,

$$
\begin{aligned}
\frac{\partial}{\partial_t} \| \hat{\boldsymbol{x}}_t^\phi - \boldsymbol{x}_t \|_2^2 &= 2 \langle \hat{\boldsymbol{x}}_t^\phi - \boldsymbol{x}_t, \frac{\partial}{\partial_t}(\hat{\boldsymbol{x}}_t^\phi - \boldsymbol{x}_t) \rangle \\
&= 2 \left( \| \hat{\boldsymbol{x}}_t^\phi - \boldsymbol{x}_t \|_2^2 + \langle \hat{\boldsymbol{x}}_t^\phi - \boldsymbol{x}_t, \boldsymbol{s}(\boldsymbol{x}_{t_{k+1}}, t_{k+1}) - \nabla \log p_t(\boldsymbol{x}_t) \rangle \right) \\
&\leq (2 + \frac{1}{h_k}) \| \hat{\boldsymbol{x}}_t^\phi - \boldsymbol{x}_t \|_2^2 + h_k \| \boldsymbol{s}(\boldsymbol{x}_{t_{k+1}}, t_{k+1}) - \nabla \log p_t(\boldsymbol{x}_t) \|_2^2.
\end{aligned}
\tag{22}
$$

By Grönwall's inequality,

$$
\begin{aligned}
&\mathbb{E}_{\boldsymbol{x}_{t_{k+1}} \sim p_{t_{k+1}}} \left[ \| \hat{\boldsymbol{x}}_{t_k}^\phi - \boldsymbol{x}_{t_k} \|_2^2 \right] \\
&\leq \exp((2 + \frac{1}{h_k})h_k) \int_{t_k}^{t_{k+1}} h_k \mathbb{E}_{\boldsymbol{x}_{t_{k+1}} \sim p_{t_{k+1}}} \left[ \| \boldsymbol{s}(\boldsymbol{x}_{t_{k+1}}, t_{k+1}) - \nabla \log p_t(\boldsymbol{x}_t) \|_2^2 \right] \mathrm{d}t \\
&\lesssim h_k \int_{t_k}^{t_{k+1}} \mathbb{E}_{\boldsymbol{x}_{t_{k+1}} \sim p_{t_{k+1}}} \left[ \| \boldsymbol{s}(\boldsymbol{x}_{t_{k+1}}, t_{k+1}) - \nabla \log p_t(\boldsymbol{x}_t) \|_2^2 \right] \mathrm{d}t.
\end{aligned}
\tag{23}
$$

We split up the error term as

$$
\begin{aligned}
&\| \boldsymbol{s}(\boldsymbol{x}_{t_{k+1}}, t_{k+1}) - \nabla \log p_t(\boldsymbol{x}_t) \|_2^2 \\
&\qquad \lesssim \| \boldsymbol{s}(\boldsymbol{x}_{t_{k+1}}, t_{k+1}) - \nabla \log p_{t_{k+1}}(\boldsymbol{x}_{t_{k+1}}) \|_2^2 + \| \nabla \log p_{t_{k+1}}(\boldsymbol{x}_{t_{k+1}}) - \nabla \log p_t(\boldsymbol{x}_t) \|_2^2.
\end{aligned}
\tag{24}
$$

By assumption 3, the first term is bounded in expectation by $\varepsilon_{\text{sc}}^2$. By Lemma 11 and $t_k \leq t \leq t_{k+1}$, the second term is bounded by

$$
\begin{aligned}
\mathbb{E}_{\boldsymbol{x}_{t_{k+1}} \sim p_{t_{k+1}}} \left[ \| \nabla \log p_{t_{k+1}}(\boldsymbol{x}_{t_{k+1}}) - \nabla \log p_t(\boldsymbol{x}_t) \|_2^2 \right] &= \mathbb{E}_{\boldsymbol{x}_{t_{k+1}} \sim p_{t_{k+1}}} \left[ \left\| \int_t^{t_{k+1}} \frac{\partial}{\partial u} \nabla \log p_u(\boldsymbol{x}_u) \mathrm{d}u \right\|_2^2 \right] \\
&\leq (t_{k+1} - t) \int_t^{t_{k+1}} \mathbb{E} \left[ \left\| \frac{\partial}{\partial u} \nabla \log p_u(\boldsymbol{x}_u) \right\|_2^2 \right] \mathrm{d}u \\
&\leq h_k \int_t^{t_{k+1}} L_s^2 d(L_s + \frac{1}{u}) du \\
&\lesssim L_s^2 d h_k^2 (L_s + \frac{1}{t_k}),
\end{aligned}
\tag{25}
$$

thus

$$
\mathbb{E}_{\boldsymbol{x}_{t_{k+1}} \sim p_{t_{k+1}}} \left[ \| \hat{\boldsymbol{x}}_{t_k}^{\phi} - \boldsymbol{x}_{t_k} \|_2^2 \right] \leq h_k^2 (L_s^2 d h_k^2 (L_s + \frac{1}{t_k}) + \varepsilon_{\text{sc}}^2),
$$

Now take it back to equation 20

$$
\begin{aligned}
E_2 &\lesssim L_f L_s^{\frac{3}{2}} d^{\frac{1}{2}} \sum_{k=1}^N h_k^2 + L_f L_s d^{\frac{1}{2}} \sum_{k=1}^N \frac{h_k^2}{t_k^{\frac{1}{2}}} + L_f \varepsilon_{\text{sc}} \sum_{k=1}^N h_k \\
&\lesssim L_f L_s^{\frac{3}{2}} d^{\frac{1}{2}} h(t_n - t_1) + L_f L_s d^{\frac{1}{2}} h(t_n - t_1)^{\frac{1}{2}} + L_f \varepsilon_{\text{sc}}(t_n - t_1)
\end{aligned}
\tag{26}
$$

as we specially designed $h_k$ for $k \in [\![1, N_1]\!]$ such that $h_k \leq \frac{t_{k+1}}{2}$, which implies $\sum_{k=1}^N \frac{h_k}{t_k^{\frac{1}{2}}}$ is a constant-factor approximation of the integral $\int_{t_1}^{t_n} \frac{1}{t^{\frac{1}{2}}} \mathrm{d}t \lesssim \sqrt{t_n - t_1}$.

Combining equations 19 and 26, we immediately get the result.

$\square$

## B.2 PROOF OF THEOREM 3 AND COROLLARY 4

**Proof of Theorem 3.** Take a couple of $(\boldsymbol{Y}, \boldsymbol{Z}) \sim \gamma(\boldsymbol{y}, \boldsymbol{z})$ where $\gamma \in \Gamma(\mu, p_{t_n})$ is a coupling between $\mu$ and $p_{t_n}$, that is,

$$
\int_{\mathbb{R}^d} \gamma(\boldsymbol{y}, \boldsymbol{z}) \mathrm{d}\boldsymbol{z} = \mu(\boldsymbol{y}),
\tag{27}
$$

$$
\int_{\mathbb{R}^d} \gamma(\boldsymbol{y}, \boldsymbol{z}) \mathrm{d}\boldsymbol{y} = p_{t_n}(\boldsymbol{z}).
\tag{28}
$$

then we have $\boldsymbol{f}_{\boldsymbol{\theta}}(\boldsymbol{Y}, t_n) \sim \boldsymbol{f}_{\boldsymbol{\theta}, t_n} \sharp \mu$, $\boldsymbol{f}^{\text{ex}}(\boldsymbol{Z}, t_n) \sim p_\delta$, thus

$$
\begin{aligned}
W_2(\boldsymbol{f}_{\boldsymbol{\theta}, t_n} \sharp \mu, p_\delta) &\leq \left( \mathbb{E}_\gamma \left[ \| \boldsymbol{f}_{\boldsymbol{\theta}}(\boldsymbol{Y}, t_n) - \boldsymbol{f}^{\text{ex}}(\boldsymbol{Z}, t_n) \|_2^2 \right] \right)^{1/2} \\
&\leq \left( \mathbb{E}_\gamma \left[ \| \boldsymbol{f}_{\boldsymbol{\theta}}(\boldsymbol{Y}, t_n) - \boldsymbol{f}_{\boldsymbol{\theta}}(\boldsymbol{Z}, t_n) \|_2^2 \right] \right)^{1/2} + \left( \mathbb{E}_\gamma \left[ \| \boldsymbol{f}_{\boldsymbol{\theta}}(\boldsymbol{Z}, t_n) - \boldsymbol{f}^{\text{ex}}(\boldsymbol{Z}, t_n) \|_2^2 \right] \right)^{1/2} \\
&\leq L_f \left( \mathbb{E}_\gamma \left[ \| \boldsymbol{Y} - \boldsymbol{Z} \|_2^2 \right] \right)^{1/2} + t_n (\varepsilon_{\text{cm}} + L_f \varepsilon_{\text{sc}} + L_f L_s^{\frac{3}{2}} d^{\frac{1}{2}} h) + t_n^{\frac{1}{2}} L_f L_s d^{\frac{1}{2}} h.
\end{aligned}
\tag{29}
$$

Note that $\gamma$ can be any coupling between $\mu$ and $p_{t_n}$, this implies

$$
W_2(\boldsymbol{f}_{\boldsymbol{\theta}, t_n} \sharp \mu, p_\delta) \leq L_f W_2(\mu, p_{t_n}) + t_n(\varepsilon_{\text{cm}} + L_f \varepsilon_{\text{sc}} + L_f L_s^{\frac{3}{2}} d^{\frac{1}{2}} h) + t_n^{\frac{1}{2}} L_f L_s d^{\frac{1}{2}} h
$$

$\square$

**Proof of Corollary 4.** We first proof that

$$W_2(\boldsymbol{f}_{\boldsymbol{\theta},T}\sharp\mathcal{N}(\mathbf{0},\boldsymbol{I}_d),p_\delta) \lesssim (d^{\frac{1}{2}}\vee \mathfrak{m})L_f e^{-T} + T(\varepsilon_{\mathrm{cm}} + L_f\varepsilon_{\mathrm{sc}} + L_f L_s^{\frac{3}{2}}d^{\frac{1}{2}}h). \tag{30}$$

This follows directly from the fact that $e^{-T}\boldsymbol{x}_0 + \sqrt{1-e^{-2T}}\boldsymbol{\xi} \sim p_T(\boldsymbol{x})$, if $\boldsymbol{\xi} \sim \mathcal{N}(\mathbf{0},\boldsymbol{I}_d)$, thus

$$W_2(\mathcal{N}(\mathbf{0},(1-e^{-2T})\boldsymbol{I}_d),p_T) \leq \left(\mathbb{E}_{p_{\mathrm{data}}}[\|e^{-T}\boldsymbol{x}_0 + (\sqrt{1-e^{-2T}}-1)\xi\|_2^2]\right)^{1/2} \lesssim (\sqrt{d}\vee\mathfrak{m})e^{-T}$$

and Theorem 3, where we taking $\mu = \mathcal{N}(\mathbf{0},\boldsymbol{I}_d)$.

The corollary then follow from a simple triangular inequality and the fact that

$$\begin{aligned}
W_2(p_\delta,p_0) &\leq \left(\mathbb{E}_{p_{\mathrm{data}}}[\|(1-e^{-\delta})\boldsymbol{x}_0 + (\sqrt{1-e^{-2\delta}})\xi\|_2^2]\right)^{1/2} \\
&\leq ((1-e^{-\delta})^2\mathfrak{m}^2 + (1-e^{-2\delta})d)^{1/2} \\
&\lesssim (\sqrt{d}\vee\mathfrak{m})\sqrt{\delta}.
\end{aligned} \tag{31}$$

$\square$

### B.3 Proofs for Corollary 5 and 6

**Proof of Corollary 5.** Take a couple of $(\boldsymbol{Y},\boldsymbol{Z}) \sim \gamma(\boldsymbol{y},\boldsymbol{z})$ where $\gamma \in \Gamma(q_{k-1},p_\delta)$, take $\boldsymbol{\xi} \sim \mathcal{N}(\mathbf{0},\boldsymbol{I}_d)$, then we have

$$\begin{aligned}
\hat{\boldsymbol{Y}} &= e^{-(t_{k_n}-\delta)}\boldsymbol{Y} + \sqrt{1-e^{-2(t_{k_n}-\delta)}}\boldsymbol{\xi} \sim \mu_k, \\
\hat{\boldsymbol{Z}} &= e^{-(t_{k_n}-\delta)}\boldsymbol{Z} + \sqrt{1-e^{-2(t_{k_n}-\delta)}}\boldsymbol{\xi} \sim p_{t_{n_k}},
\end{aligned} \tag{32}$$

The statement follows from the fact that

$$\begin{aligned}
W_2(\boldsymbol{f}_{\boldsymbol{\theta},t_{n_k}}\sharp\mu_k,p_\delta) &\lesssim L_f W_2(\mu_k,p_{t_{n_k}}) + t_n(\varepsilon_{\mathrm{cm}} + L_f\varepsilon_{\mathrm{sc}} + L_f L_s^{\frac{3}{2}}d^{\frac{1}{2}}h) + t_n^{\frac{1}{2}}L_f L_s d^{\frac{1}{2}}h \\
&\leq L_f(\mathbb{E}_\gamma\|\hat{\boldsymbol{Y}}-\hat{\boldsymbol{Z}}\|_2^2)^{1/2} + t_n(\varepsilon_{\mathrm{cm}} + L_f\varepsilon_{\mathrm{sc}} + L_f L_s^{\frac{3}{2}}d^{\frac{1}{2}}h) + t_n^{\frac{1}{2}}L_f L_s d^{\frac{1}{2}}h \\
&= L_f e^{-(t_{k_n}-\delta)}(\mathbb{E}_\gamma\|\boldsymbol{Y}-\boldsymbol{Z}\|_2^2)^{1/2} + t_n(\varepsilon_{\mathrm{cm}} + L_f\varepsilon_{\mathrm{sc}} + L_f L_s^{\frac{3}{2}}d^{\frac{1}{2}}h) + t_n^{\frac{1}{2}}L_f L_s d^{\frac{1}{2}}h
\end{aligned} \tag{33}$$

and $\gamma$ can be arbitrary coupling between $q_{k-1}$ and $p_\delta$, which means

$$W_2(q_k,p_\delta) \lesssim L_f e^{-(t_{n_k}-\delta)}W_2(q_{k-1},p_\delta) + t_{n_k}(\varepsilon_{\mathrm{cm}} + L_f\varepsilon_{\mathrm{sc}} + L_f L_s^{\frac{3}{2}}d^{\frac{1}{2}}h) + t_{n_k}^{\frac{1}{2}}L_f L_s d^{\frac{1}{2}}h.$$

$\square$

**Proof of Corollary 6.** For the first statement, fix $n_k \equiv \hat{n}$, note that $L_s \geq 1, t_{\hat{n}} \geq \log(2L_f) + \delta$, according to the proof of Corollary 5, we may assume $C$ is the constant factor such that

$$W_2(q_k,p_\delta) \leq L_f e^{-(t_{\hat{n}}-\delta)}W_2(q_{k-1},p_\delta) + Ct_{\hat{n}}(\varepsilon_{\mathrm{cm}} + L_f\varepsilon_{\mathrm{sc}} + L_f L_s^{\frac{3}{2}}d^{\frac{1}{2}}h).$$

where we omit the term with $t_{\hat{n}}^{\frac{1}{2}}$ as it is controlled by the terms with $t_{\hat{n}}$. Denote

$$D = C(\varepsilon_{\mathrm{cm}} + L_f\varepsilon_{\mathrm{sc}} + L_f L_s^{\frac{3}{2}}d^{\frac{1}{2}}h) \text{ and } \mathcal{E}_k = W_2(q_k,p_{\mathrm{data}})$$

for short, we have

$$\mathcal{E}_k \leq L_f e^{-(t_{\hat{n}}-\delta)}\mathcal{E}_{k-1} + t_{\hat{n}}D,$$

which means

$$\mathcal{E}_k - \frac{t_{\hat{n}}D}{1 - L_f e^{-(t_{\hat{n}}-\delta)}} \leq L_f e^{-(t_{\hat{n}}-\delta)}\left(\mathcal{E}_{k-1} - \frac{t_{\hat{n}}D}{1 - L_f e^{-(t_{\hat{n}}-\delta)}}\right),$$

thus,

1. if $\mathcal{E}_{k-1} \leq \frac{t_{\hat{n}}D}{1-L_f e^{-(t_{\hat{n}}-\delta)}}$, $\mathcal{E}_K \leq \frac{t_{\hat{n}}D}{1-L_f e^{-(t_{\hat{n}}-\delta)}}$ for all $K \geq k$;

2. if $\mathcal{E}_{k-1} > \frac{t_{\hat{n}} D}{1 - L_f e^{-(t_{\hat{n}} - \delta)}}$,

$$\mathcal{E}_k \le \frac{t_{\hat{n}} D}{1 - L_f e^{-(t_{\hat{n}} - \delta)}} + \left( L_f e^{-(t_{\hat{n}} - \delta)} \right)^{k-1} \left( \mathcal{E}_1 - \frac{t_{\hat{n}} D}{1 - L_f e^{-(t_{\hat{n}} - \delta)}} \right). \tag{34}$$

These observation shows that $\mathcal{E}_k$ is exponentially upper-bounded by $\frac{t_{\hat{n}} D}{1 - L_f e^{-(t_{\hat{n}} - \delta)}}$, and such an upper bound can be further minimized over $\hat{n} \in [\![1, N]\!]$: in fact if we take $t_{\hat{n}} \approx \log(2L_f) + \delta$, then

$$\frac{t_{\hat{n}} D}{1 - L_f e^{-(t_{\hat{n}} - \delta)}} \approx 2 \log(2L_f) D = O(\log(L_f) D).$$

together with $L_f e^{-(t_{\hat{n}} - \delta)} \approx \frac{1}{2}$, $\mathcal{E}_1 = O((d^{\frac{1}{2}} \vee \mathfrak{m}) L_f e^{-T} + T(\varepsilon_{\mathrm{cm}} + L_f \varepsilon_{\mathrm{sc}} + L_f L_s^{\frac{3}{2}} d^{\frac{1}{2}} h))$, this completes the proof of the first statement.

The second statement follows from the fact that

$$\begin{aligned} W_2(p_\delta, p_0) &\le \left( \mathbb{E}_{p_{\mathrm{data}}}[\|(1 - e^{-\delta}) \boldsymbol{x}_0 + (\sqrt{1 - e^{-2\delta}}) \xi\|_2^2] \right)^{1/2} \\ &\le ((1 - e^{-\delta})^2 \mathfrak{m}^2 + (1 - e^{-2\delta}) d)^{1/2} \\ &\lesssim (\sqrt{d} \vee \mathfrak{m}) \sqrt{\delta}. \end{aligned} \tag{35}$$

and a simple use of triangular inequality: $W_2(q_k, p_0) \le W_2(q_k, p_\delta) + W_2(p_\delta, p_0)$. $\qquad\square$

## B.4 Proof of Lemma 7 and Corollary 8

**Proof of Lemma 7.** Let $\mu_{\boldsymbol{x}, \sigma^2}$ be the density $\mu(\mathrm{d}\boldsymbol{u})$ weighted with the gaussian $\psi_{\sigma^2}(\boldsymbol{u} - \boldsymbol{x}) \sim e^{-\frac{\|\boldsymbol{x} - \boldsymbol{u}\|_2^2}{2\sigma^2}}$, that is,

$$\mu_{\boldsymbol{x}, \sigma^2}(\mathrm{d}\boldsymbol{u}) = \frac{e^{-\frac{\|\boldsymbol{x} - \boldsymbol{u}\|_2^2}{2\sigma^2}} \mu(\mathrm{d}\boldsymbol{u})}{\int_{\mathbb{R}^d} e^{-\frac{\|\boldsymbol{x} - \tilde{\boldsymbol{u}}\|_2^2}{2\sigma^2}} \mu(\mathrm{d}\tilde{\boldsymbol{u}})}.$$

Note that

$$\nabla \log(\mu * \psi_{\sigma^2}(\boldsymbol{x})) = \frac{\nabla \int_{\mathbb{R}^d} e^{-\frac{\|\boldsymbol{x} - \boldsymbol{u}\|_2^2}{2\sigma^2}} \mu(\mathrm{d}\boldsymbol{u})}{\int_{\mathbb{R}^d} e^{-\frac{\|\boldsymbol{x} - \boldsymbol{u}\|_2^2}{2\sigma^2}} \mu(\mathrm{d}\boldsymbol{u})} = \frac{\int_{\mathbb{R}^d} -\frac{\boldsymbol{x} - \boldsymbol{u}}{\sigma^2} e^{-\frac{\|\boldsymbol{x} - \boldsymbol{u}\|_2^2}{2\sigma^2}} \mu(\mathrm{d}\boldsymbol{u})}{\int_{\mathbb{R}^d} e^{-\frac{\|\boldsymbol{x} - \boldsymbol{u}\|_2^2}{2\sigma^2}} \mu(\mathrm{d}\boldsymbol{u})} = -\frac{1}{\sigma^2} \mathbb{E}_{\mu_{\boldsymbol{x}, \sigma^2}}[\boldsymbol{x} - \boldsymbol{u}],$$

$$\begin{aligned} \nabla^2 \log(\mu * \psi_{\sigma^2}(\boldsymbol{x})) &= \frac{\nabla \otimes \int_{\mathbb{R}^d} -\frac{\boldsymbol{x} - \boldsymbol{u}}{\sigma^2} e^{-\frac{\|\boldsymbol{x} - \boldsymbol{u}\|_2^2}{2\sigma^2}} \mu(\mathrm{d}\boldsymbol{u})}{\int_{\mathbb{R}^d} e^{-\frac{\|\boldsymbol{x} - \boldsymbol{u}\|_2^2}{2\sigma^2}} \mu(\mathrm{d}\boldsymbol{u})} - \left( \frac{\int_{\mathbb{R}^d} -\frac{\boldsymbol{x} - \boldsymbol{u}}{\sigma^2} e^{-\frac{\|\boldsymbol{x} - \boldsymbol{u}\|_2^2}{2\sigma^2}} \mu(\mathrm{d}\boldsymbol{u})}{\int_{\mathbb{R}^d} e^{-\frac{\|\boldsymbol{x} - \boldsymbol{u}\|_2^2}{2\sigma^2}} \mu(\mathrm{d}\boldsymbol{u})} \right)^{\otimes 2} \\ &= -\frac{1}{\sigma^2} \boldsymbol{I}_d + \frac{\int_{\mathbb{R}^d} \left( \frac{\boldsymbol{x} - \boldsymbol{u}}{\sigma^2} \right)^{\otimes 2} e^{-\frac{\|\boldsymbol{x} - \boldsymbol{u}\|_2^2}{2\sigma^2}} \mu(\mathrm{d}\boldsymbol{u})}{\int_{\mathbb{R}^d} e^{-\frac{\|\boldsymbol{x} - \boldsymbol{u}\|_2^2}{2\sigma^2}} \mu(\mathrm{d}\boldsymbol{u})} - \left( \frac{\int_{\mathbb{R}^d} -\frac{\boldsymbol{x} - \boldsymbol{u}}{\sigma^2} e^{-\frac{\|\boldsymbol{x} - \boldsymbol{u}\|_2^2}{2\sigma^2}} \mu(\mathrm{d}\boldsymbol{u})}{\int_{\mathbb{R}^d} e^{-\frac{\|\boldsymbol{x} - \boldsymbol{u}\|_2^2}{2\sigma^2}} \mu(\mathrm{d}\boldsymbol{u})} \right)^{\otimes 2} \\ &= \frac{1}{\sigma^4} \int_{\mathbb{R}^d} (\boldsymbol{u} - \mathbb{E}_{\mu_{\boldsymbol{x}, \sigma^2}}[\boldsymbol{u}]) \otimes (\boldsymbol{u} - \mathbb{E}_{\mu_{\boldsymbol{x}, \sigma^2}}[\boldsymbol{u}]) \mu_{\boldsymbol{x}, \sigma^2}(\mathrm{d}\boldsymbol{u}) - \frac{1}{\sigma^2} \boldsymbol{I}_d \end{aligned} \tag{36}$$

where for any vector $\boldsymbol{y} \in \mathbb{R}^d$, we denote $\boldsymbol{y}^{\otimes 2} = \boldsymbol{y} \otimes \boldsymbol{y} = \boldsymbol{y}\boldsymbol{y}^T \in \mathbb{R}^{d \times d}$ as a matrix and denote $\nabla \otimes \boldsymbol{f}(\boldsymbol{x})$ as the Jacobbian matrix $[\frac{\partial f_i(\boldsymbol{x})}{\partial x_j}]_{1 \le i, j \le d}$

Note that if $\mu$ is bounded on a set of radius $R$, so as $\mu_{\boldsymbol{x}, \sigma^2}$, then the covariance of $\mu_{\boldsymbol{x}, \sigma^2}$ is bounded by $R^2$ in operator norm.

Now we take $\mu(\boldsymbol{u}) = e^{dt} p_0(e^t \boldsymbol{u})$, which is bounded on a set of radius $e^{-t} R$. Take $\sigma^2 = 1 - e^{-2t}$, we have $\mu * \psi_{\sigma^2}(\boldsymbol{x}) = p_t(\boldsymbol{x})$, thus

$$\|\nabla^2 \log p_t(\boldsymbol{x})\|_{\mathrm{op}} \le \frac{e^{-2t} R^2}{(1 - e^{-2t})^2} + \frac{1}{1 - e^{-2t}}.$$

$\qquad\square$

**Proof of Corollary 8.** Note that supp $p_{\text{data}} \subseteq B(\mathbf{0}, R)$ implies $\mathbb{E}_{\boldsymbol{x}_0 \sim p_{\text{data}}}[\|\boldsymbol{x}_0\|_2^2] \lesssim R^2$. Taking $\delta \asymp \frac{\varepsilon^2}{R^2 \vee d}$, according to Lemma 7, we have

$$\|\nabla^2 \log p_t(\boldsymbol{x})\|_{\text{op}} \leq \frac{e^{-2t}R^2}{(1 - e^{-2t})^2} + \frac{1}{1 - e^{-2t}} \lesssim \frac{1}{t} \vee \frac{R^2}{t^2} \lesssim \frac{R^2(R^2 \vee d)^2}{\varepsilon^4}, \forall t \geq \delta$$

and thus $\nabla \log p_t(\boldsymbol{x})$ satisfies Assumption 2 with $L_s \asymp \frac{R^2(R^2 \vee d)^2}{\varepsilon^4}$ for $t \geq \delta$. Now according to Corollary 4, we have that

$$W_2(\boldsymbol{f}_{\boldsymbol{\theta},T} \sharp \mathcal{N}(\mathbf{0}, \boldsymbol{I}_d), p_{\text{data}}) \lesssim (d^{\frac{1}{2}} \vee R)L_f e^{-T} + T(\varepsilon_{\text{cm}} + L_f \varepsilon_{\text{sc}} + \frac{L_f d^{\frac{1}{2}} R^3 (R^2 \vee d)^3 h}{\varepsilon^6}) + \varepsilon,$$

thus if we take $h = O(\frac{\varepsilon^7}{d^{1/2} R^3 (R^6 \vee d^3) L_f T})$, $\varepsilon_{\text{cm}} = O(\frac{\varepsilon}{T})$, $\varepsilon_{\text{sc}} = O(\frac{\varepsilon}{L_f T})$, $T = O(\log(\frac{L_f(\sqrt{d} \vee R)}{\varepsilon}))$, we can guarantee $W_2(\boldsymbol{f}_{\boldsymbol{\theta},T} \sharp \mathcal{N}(\mathbf{0}, \boldsymbol{I}_d), p_{\text{data}}) \lesssim \varepsilon$.

Similarly, according to Corollary 6, we have

$$W_2(q_k, p_{\text{data}}) \lesssim (\log(L_f) + 2^{-k}T)(\varepsilon_{\text{cm}} + L_f \varepsilon_{\text{sc}} + \frac{L_f d^{\frac{1}{2}} R^3 (R^2 \vee d)^3 h}{\varepsilon^6}) + 2^{-k}(d^{\frac{1}{2}} \vee \mathfrak{m})L_f e^{-T} + \varepsilon,$$

thus if we take $h = O(\frac{\varepsilon^7}{d^{1/2} R^3 (R^6 \vee d^3) L_f \log(L_f)})$, $\varepsilon_{\text{cm}} = O(\frac{\varepsilon}{\log(L_f)})$, $\varepsilon_{\text{sc}} = O(\frac{\varepsilon}{L_f \log(L_f)})$, $k = O(\log(T \vee (\frac{(d \vee \mathfrak{m}^2)L_f L_s}{\varepsilon})))$, we can guarantee $W_2(q_k, p_{\text{data}}) \lesssim \varepsilon$. $\qquad \square$

### B.5 PROOF OF COROLLARY 9

Before we prove the corollary 9, we first prove the following lemma, which shows that TV error can be bounded after a small time OU regularization.

**Lemma 12.** *For any two distribution $p$ and $q$, running the OU process 16 for $p, q$ individually with time $\tau > 0$, the following TV distance bound holds,*

$$TV(pP_{OU}^{\tau}, qP_{OU}^{\tau}) \lesssim \frac{1}{\sqrt{\tau}} W_1(p, q) \leq \frac{1}{\sqrt{\tau}} W_2(p, q)$$

*Proof.* Denote $\psi_{\sigma^2}(\boldsymbol{y})$ as the density function to the normal distribution $\mathcal{N}(\mathbf{0}, \sigma^2 \boldsymbol{I}_d)$. We write the $TV(pP_{\text{OU}}^{\tau}, qP_{\text{OU}}^{\tau})$ into integral form as:

$$TV(pP_{\text{OU}}^{\tau}, qP_{\text{OU}}^{\tau}) = \frac{1}{2} \int_{\mathbb{R}^d} |(pP_{\text{OU}}^{\tau})(\boldsymbol{x}) - (qP_{\text{OU}}^{\tau})(\boldsymbol{x})| \mathrm{d}\boldsymbol{x}$$
$$= \frac{1}{2} \int_{\mathbb{R}^d} \left| \int_{\mathbb{R}^d} p(\boldsymbol{y})\psi_{1-e^{-2\tau}}(\boldsymbol{x} - e^{-\tau}\boldsymbol{y})\mathrm{d}\boldsymbol{y} - \int_{\mathbb{R}^d} q(\boldsymbol{z})\psi_{1-e^{-2\tau}}(\boldsymbol{x} - e^{-\tau}\boldsymbol{z})\mathrm{d}\boldsymbol{z} \right| \mathrm{d}\boldsymbol{x}.$$
$$\tag{37}$$

Taking a coupling $\gamma \in \Gamma(p, q)$,

$$\int_{\mathbb{R}^d} \gamma(\boldsymbol{y}, \boldsymbol{z})\mathrm{d}\boldsymbol{z} = p(\boldsymbol{y}),$$
$$\int_{\mathbb{R}^d} \gamma(\boldsymbol{y}, \boldsymbol{z})\mathrm{d}\boldsymbol{y} = q(\boldsymbol{z}),$$
$$\tag{38}$$

we have

$$
\begin{aligned}
\mathrm{TV}(pP_{\mathrm{OU}}^\tau, qP_{\mathrm{OU}}^\tau) &= \frac{1}{2}\int_{\mathbb{R}^d}\left|\int_{\mathbb{R}^{d\times d}}\gamma(\boldsymbol{y},\boldsymbol{z})[\psi_{1-e^{-2\tau}}(\boldsymbol{x}-e^{-\tau}\boldsymbol{y})-\psi_{1-e^{-2\tau}}(\boldsymbol{x}-e^{-\tau}\boldsymbol{z})]\mathrm{d}\boldsymbol{y}\mathrm{d}\boldsymbol{z}\right|\mathrm{d}\boldsymbol{x}. \\
&\leq \frac{1}{2}\int_{\mathbb{R}^d}\int_{\mathbb{R}^{d\times d}}\gamma(\boldsymbol{y},\boldsymbol{z})\left|\psi_{1-e^{-2\tau}}(\boldsymbol{x}-e^{-\tau}\boldsymbol{y})-\psi_{1-e^{-2\tau}}(\boldsymbol{x}-e^{-\tau}\boldsymbol{z})\right|\mathrm{d}\boldsymbol{y}\mathrm{d}\boldsymbol{z}\mathrm{d}\boldsymbol{x} \\
&= \int_{\mathbb{R}^{d\times d}}\gamma(\boldsymbol{y},\boldsymbol{z})\left(\frac{1}{2}\int_{\mathbb{R}^d}\left|\psi_{1-e^{-2\tau}}(\boldsymbol{x}-e^{-\tau}\boldsymbol{y})-\psi_{1-e^{-2\tau}}(\boldsymbol{x}-e^{-\tau}\boldsymbol{z})\right|\mathrm{d}\boldsymbol{x}\right)\mathrm{d}\boldsymbol{y}\mathrm{d}\boldsymbol{z} \\
&= \int_{\mathbb{R}^{d\times d}}\gamma(\boldsymbol{y},\boldsymbol{z})\mathrm{TV}\left(\psi_{1-e^{-2\tau}}(\cdot-e^{-\tau}\boldsymbol{y}),\psi_{1-e^{-2\tau}}(\cdot-e^{-\tau}\boldsymbol{z})\right)\mathrm{d}\boldsymbol{y}\mathrm{d}\boldsymbol{z} \\
&\leq \int_{\mathbb{R}^{d\times d}}\gamma(\boldsymbol{y},\boldsymbol{z})\sqrt{\frac{1}{2}\mathrm{KL}\left(\psi_{1-e^{-2\tau}}(\cdot-e^{-\tau}\boldsymbol{y})\middle\|\psi_{1-e^{-2\tau}}(\cdot-e^{-\tau}\boldsymbol{z})\right)}\mathrm{d}\boldsymbol{y}\mathrm{d}\boldsymbol{z} \\
&= \int_{\mathbb{R}^{d\times d}}\gamma(\boldsymbol{y},\boldsymbol{z})\frac{1}{2}\sqrt{\frac{e^{-2\tau}}{1-e^{-2\tau}}\|\boldsymbol{y}-\boldsymbol{z}\|_2^2}\mathrm{d}\boldsymbol{y}\mathrm{d}\boldsymbol{z} \\
&= \frac{1}{2\sqrt{e^{2\tau}-1}}\int_{\mathbb{R}^{d\times d}}\gamma(\boldsymbol{y},\boldsymbol{z})\|\boldsymbol{y}-\boldsymbol{z}\|_2\mathrm{d}\boldsymbol{y}\mathrm{d}\boldsymbol{z}, \quad (39)
\end{aligned}
$$

where for the second inequality we use the fact that $\mathrm{TV}(u,v) \leq \sqrt{\frac{1}{2}\mathrm{KL}(u\|v)}$, and the next equality we use the formula of KL divergence between two Gaussian distribution. Noting $\frac{1}{2\sqrt{e^{2\tau}-1}} \leq \frac{1}{2\sqrt{2\tau}}$, and taking $\gamma$ over all coupling $\Gamma(p,q)$, we have

$$
\mathrm{TV}(pP_{\mathrm{OU}}^\tau, qP_{\mathrm{OU}}^\tau) \lesssim \frac{1}{\sqrt{\tau}}W_1(p,q) \leq \frac{1}{\sqrt{\tau}}W_2(p,q)
$$

$\square$

**Proof of Corollary 9.** According to the triangular inequality, Lemma 12 and equation 12,

$$
\begin{aligned}
\mathrm{TV}(q_1P_{\mathrm{OU}}^\delta, p_{\mathrm{data}}) &\leq \mathrm{TV}(q_1P_{\mathrm{OU}}^\delta, p_\delta P_{\mathrm{OU}}^\delta) + \mathrm{TV}(p_\delta P_{\mathrm{OU}}^\delta, p_{\mathrm{data}}) \\
&\lesssim \frac{1}{\sqrt{\delta}}W_2(q_1, p_\delta) + \mathrm{TV}(p_{2\delta}, p_{\mathrm{data}}) \\
&\lesssim \frac{1}{\sqrt{\delta}}\left((d^{\frac{1}{2}}\vee\mathfrak{m})L_fe^{-T} + T(\varepsilon_{\mathrm{cm}} + L_f\varepsilon_{\mathrm{sc}} + L_fL_s^{\frac{3}{2}}d^{\frac{1}{2}}h)\right) + \mathrm{TV}(p_{2\delta}, p_{\mathrm{data}}).
\end{aligned}
$$
$$(40)$$

Note that if we take $\delta \asymp \frac{\varepsilon^2}{L_s^2(d\vee\mathfrak{m}^2)}$, then by Lemma 6.4, Lee et al. (2022b), $\mathrm{TV}(p_{2\delta}, p_{\mathrm{data}}) \leq \varepsilon$, this concludes that

$$
\mathrm{TV}(q_1P_{\mathrm{OU}}^\delta, p_{\mathrm{data}}) \lesssim \frac{L_s(d^{\frac{1}{2}}\vee\mathfrak{m})}{\varepsilon}[(\log L_f + \frac{T}{2^k})(\varepsilon_{\mathrm{cm}} + L_f\varepsilon_{\mathrm{sc}} + L_fL_s^{\frac{3}{2}}d^{\frac{1}{2}}h) + \frac{(d^{\frac{1}{2}}\vee\mathfrak{m})L_f}{2^ke^T}] + \varepsilon.
$$

Similarly for $q_k$, if we take $\delta \asymp \frac{\varepsilon^2}{L_s^2(d\vee\mathfrak{m}^2)}$

$$
\begin{aligned}
&\mathrm{TV}(q_kP_{\mathrm{OU}}^\delta, p_{\mathrm{data}}) \quad\quad\quad\quad\quad\quad\quad\quad\quad\quad\quad\quad\quad\quad\quad\quad\quad\quad\quad\quad (41) \\
&\leq \mathrm{TV}(q_kP_{\mathrm{OU}}^\delta, p_\delta P_{\mathrm{OU}}^\delta) + \mathrm{TV}(p_\delta P_{\mathrm{OU}}^\delta, p_{\mathrm{data}}) \\
&\lesssim \frac{L_s(d^{\frac{1}{2}}\vee\mathfrak{m})}{\varepsilon}[(\log L_f + \frac{T}{2^k})(\varepsilon_{\mathrm{cm}} + L_f\varepsilon_{\mathrm{sc}} + L_fL_s^{\frac{3}{2}}d^{\frac{1}{2}}h) + \frac{(d^{\frac{1}{2}}\vee\mathfrak{m})L_f}{2^ke^T}] + \varepsilon. \quad (42)
\end{aligned}
$$

$\square$

### B.6 PROOF OF COROLLARY 10

We first introduce the following lemma which originally comes from Theorem 5 in Chen et al. (2023a).

**Lemma 13** (Underdamped corrector, Theorem 5 in Chen et al. (2023a)). *Denote the Markov kernel $\hat{P}_{ULMC}$ to be defined by the equation 17. Suppose $\gamma \asymp L_s$, $\nabla U$ is $L_s$-Lipschitz, $p \propto \exp(-U)$, and $\mathbb{E}_{\boldsymbol{x} \sim q}\|\boldsymbol{s}(\boldsymbol{x}) - (-\nabla U(\boldsymbol{x}))\|_2^2 \leq \varepsilon_{sc}^2$. Denote $\boldsymbol{p} := p \otimes \mathcal{N}(\boldsymbol{0}, \boldsymbol{I}_d)$, and $\boldsymbol{q} := q \otimes \mathcal{N}(\boldsymbol{0}, \boldsymbol{I}_d)$. For any $T_{corr} := N\tau \lesssim 1/\sqrt{L_s}$,*

$$TV(\boldsymbol{q}\hat{P}_{ULMC}^N, \boldsymbol{p}) \lesssim \frac{W_2(q,p)}{L_s^{1/4}T_{corr}^{3/2}} + \frac{\varepsilon_{sc}T_{corr}^{1/2}}{L_s^{1/4}} + L_s^{3/4}T_{corr}^{1/2}d^{1/2}\tau.$$

*In particular, for $T_{corr} \asymp 1/\sqrt{L_s}$,*

$$TV(\boldsymbol{q}\hat{P}_{ULMC}^N, \boldsymbol{p}) \lesssim \sqrt{L_s}W_2(q,p) + \varepsilon_{sc}/\sqrt{L_s} + \sqrt{L_s}d\tau.$$

Under the Lemma 13, we can immediately get the following result,

**Proof of Corollary 10.** Given any distribution $q$ on $\mathbb{R}^d$, we write $q\hat{P}_{\text{ULMC}}$ to denote the projection onto the $\boldsymbol{z}$-coordinates of $\boldsymbol{q}\hat{P}_{\text{ULMC}}$. Now according to 13, we have

$$\begin{aligned}
\text{TV}(q_1\hat{P}_{\text{ULMC}}^N, p_{\text{data}}) &\leq \text{TV}(q_1\hat{P}_{\text{ULMC}}^N, p_\delta) + \text{TV}(p_\delta, p_{\text{data}}) \\
&\lesssim \sqrt{L_s}W_2(q_1, p_\delta) + \varepsilon_{\text{sc}}/\sqrt{L_s} + \sqrt{L_s}d\tau + \text{TV}(p_\delta, p_{\text{data}}).
\end{aligned} \quad (43)$$

According to equation 12, we have

$$\begin{aligned}
&\text{TV}(q_1\hat{P}_{\text{ULMC}}^N, p_{\text{data}}) \\
&\lesssim (d^{\frac{1}{2}} \vee \mathfrak{m})L_f L_s^{\frac{1}{2}}e^{-T} + TL_s^{\frac{1}{2}}(\varepsilon_{\text{cm}} + L_f\varepsilon_{\text{sc}} + L_f L_s^{\frac{3}{2}}d^{\frac{1}{2}}h) + L_s^{-\frac{1}{2}}\varepsilon_{\text{sc}} + L_s^{\frac{1}{2}}d^{\frac{1}{2}}\tau + \varepsilon.
\end{aligned} \quad (44)$$

Similarly, we have

$$\begin{aligned}
&\text{TV}(q_k\hat{P}_{\text{ULMC}}^N, p_{\text{data}}) \\
&\lesssim (\log(L_f) + \frac{T}{2^k})L_s^{\frac{1}{2}}(\varepsilon_{\text{cm}} + L_f\varepsilon_{\text{sc}} + L_f L_s^{\frac{3}{2}}d^{\frac{1}{2}}h) + \frac{(d^{\frac{1}{2}} \vee \mathfrak{m})L_s^{\frac{1}{2}}L_f}{2^k e^T} + L_s^{-\frac{1}{2}}\varepsilon_{\text{sc}} + L_s^{\frac{1}{2}}d^{\frac{1}{2}}\tau + \varepsilon.
\end{aligned} \quad (45)$$

$\square$

## C    Additional proofs

**Lemma 14.** *Assuming 3 and 4, then $\hat{\boldsymbol{s}}_{\boldsymbol{\theta}}(\boldsymbol{x}) := \frac{\boldsymbol{f}_{\boldsymbol{\theta}}(\boldsymbol{x}, t_2) - e^{h_1}\boldsymbol{x}}{e^{h_1} - 1}$ is a score model approximating $\nabla \log p_{t_2}(\boldsymbol{x})$ with $L^2$ error $\varepsilon_{cm} + \varepsilon_{sc}$.*

*Proof.* Letting $h_1 = t_2 - t_1 = t_2 - \delta$, the consistency loss assumption 4 ensures

$$\mathbb{E}_{\boldsymbol{x}_{t_2} \sim p_{t_2}}\left[\|\boldsymbol{f}_{\boldsymbol{\theta}}(\boldsymbol{x}_{t_2}, t_2) - \boldsymbol{f}_{\boldsymbol{\theta}}(\hat{\boldsymbol{x}}_{t_1}^\phi, t_1)\|_2^2\right] \leq \varepsilon_{\text{cm}}^2 h_1^2,$$

where we have $\boldsymbol{f}_{\boldsymbol{\theta}}(\hat{\boldsymbol{x}}_{t_1}^\phi, t_1) = \hat{\boldsymbol{x}}_{t_1}^\phi = e^{h_1}\boldsymbol{x}_{t_2} + (e^{h_1} - 1)\boldsymbol{s}_\phi(\boldsymbol{x}_{t_2}, t_2)$, thus

$$\mathbb{E}_{\boldsymbol{x}_{t_2} \sim p_{t_2}}\left[\left\|\frac{\boldsymbol{f}_{\boldsymbol{\theta}}(\boldsymbol{x}_{t_2}, t_2) - e^{h_1}\boldsymbol{x}_{t_2}}{e^{h_1} - 1} - \boldsymbol{s}_\phi(\boldsymbol{x}_{t_2}, t_2)\right\|_2^2\right] \leq \varepsilon_{\text{cm}}^2 \frac{h_1^2}{(e^{h_1} - 1)^2} \leq \varepsilon_{\text{cm}}^2,$$

$$\begin{aligned}
&\mathbb{E}_{\boldsymbol{x}_{t_2} \sim p_{t_2}}\left[\left\|\frac{\boldsymbol{f}_{\boldsymbol{\theta}}(\boldsymbol{x}_{t_2}, t_2) - e^{h_1}\boldsymbol{x}_{t_2}}{e^{h_1} - 1} - \nabla \log p_{t_2}(\boldsymbol{x})\right\|_2^2\right] \\
&\leq \mathbb{E}_{\boldsymbol{x}_{t_2} \sim p_{t_2}}\left[\left(\left\|\frac{\boldsymbol{f}_{\boldsymbol{\theta}}(\boldsymbol{x}_{t_2}, t_2) - e^{h_1}\boldsymbol{x}_{t_2}}{e^{h_1} - 1} - \boldsymbol{s}_\phi(\boldsymbol{x}_{t_2}, t_2)\right\|_2 + \left\|\boldsymbol{s}_\phi(\boldsymbol{x}_{t_2}, t_2) - \nabla \log p_{t_2}(\boldsymbol{x})\right\|_2\right)^2\right] \\
&\leq (\varepsilon_{\text{cm}} + \varepsilon_{\text{sc}})^2.
\end{aligned}$$

$\square$

**Lemma 15.** *Under the OU scheduler 5, let $\Delta t := \max_{1 \leq n \leq N-1}(t_{n+1} - t_n)$. Assume $\boldsymbol{f}_{\boldsymbol{\theta}^-}$ is twice continuously differentiable with bounded second derivatives, and $\mathbb{E}[\|\nabla \log p_{t_n}(\boldsymbol{x}_{t_n})\|_2^2] < \infty$. Assume the CD objective 8 is defined with the exponential integrator 9 and exact score model $\boldsymbol{s}_{\boldsymbol{\phi}}(\boldsymbol{x}, t) = \nabla \log p_t(\boldsymbol{x})$, then we can define the CT objective as*

$$\mathcal{L}_{CT}^N(\boldsymbol{\theta}, \boldsymbol{\theta}^-) := \mathbb{E}[\|\boldsymbol{f}_{\boldsymbol{\theta}}(e^{-t_{n+1}}\boldsymbol{x}_0 + \sqrt{1 - e^{-2t_{n+1}}}\boldsymbol{z}, t_{n+1}) - \boldsymbol{f}_{\boldsymbol{\theta}^-}(e^{-t_n}\boldsymbol{x}_0 + \frac{1 - e^{-(t_n + t_{n+1})}}{\sqrt{1 - e^{-2t_{n+1}}}}\boldsymbol{z}, t_n)\|_2^2],$$
(46)

*where $\boldsymbol{x}_0 \sim p_{\text{data}}$, $\boldsymbol{z} \sim \mathcal{N}(\boldsymbol{0}, \boldsymbol{I}_d)$. Then we have*

$$\frac{\partial}{\partial \boldsymbol{\theta}} \mathcal{L}_{CT}^N(\boldsymbol{\theta}, \boldsymbol{\theta}^-) = \frac{\partial}{\partial \boldsymbol{\theta}} \mathcal{L}_{CD}^N(\boldsymbol{\theta}, \boldsymbol{\theta}^-) + O((\Delta t)^2)$$

*Proof.* We first prove that, if $\boldsymbol{x}_0 \sim p_{\text{data}}$, $\boldsymbol{z} \sim \mathcal{N}(\boldsymbol{0}, \boldsymbol{I}_d)$, $\boldsymbol{x}_t = e^{-t}\boldsymbol{x}_0 + \sqrt{1 - e^{-2t}}\boldsymbol{z}$, then we have $\nabla \log p_t(\boldsymbol{x}) = -\mathbb{E}[\frac{\boldsymbol{z}}{\sqrt{1 - e^{-2t}}}|\boldsymbol{x}_t]$. This comes from the fact that

$$\begin{aligned}
\nabla \log p_t(\boldsymbol{x}_t) &= \frac{\int_{\mathbb{R}^d} p_{\text{data}}(\boldsymbol{x}_0) \nabla_{\boldsymbol{x}_t} p(\boldsymbol{x}_t|\boldsymbol{x}_0) \mathrm{d}\boldsymbol{x}_0}{\int_{\mathbb{R}^d} p_{\text{data}}(\boldsymbol{x}_0) p(\boldsymbol{x}_t|\boldsymbol{x}_0) \mathrm{d}\boldsymbol{x}_0} \\
&= \frac{\int_{\mathbb{R}^d} p_{\text{data}}(\boldsymbol{x}_0) p(\boldsymbol{x}_t|\boldsymbol{x}_0) \nabla_{\boldsymbol{x}_t} \log p(\boldsymbol{x}_t|\boldsymbol{x}_0) \mathrm{d}\boldsymbol{x}_0}{p_t(\boldsymbol{x}_t)} \\
&= \int_{\mathbb{R}^d} \frac{p_{\text{data}}(\boldsymbol{x}_0) p(\boldsymbol{x}_t|\boldsymbol{x}_0)}{p_t(\boldsymbol{x}_t)} \nabla_{\boldsymbol{x}_t} \log p(\boldsymbol{x}_t|\boldsymbol{x}_0) \mathrm{d}\boldsymbol{x}_0 \\
&= \int_{\mathbb{R}^d} p(\boldsymbol{x}_0|\boldsymbol{x}_t) \nabla_{\boldsymbol{x}_t} \log p(\boldsymbol{x}_t|\boldsymbol{x}_0) \mathrm{d}\boldsymbol{x}_0 \\
&= \mathbb{E}[\nabla_{\boldsymbol{x}_t} \log p(\boldsymbol{x}_t|\boldsymbol{x}_0)|\boldsymbol{x}_t],
\end{aligned}$$

and $p(\boldsymbol{x}_t|\boldsymbol{x}_0) \sim e^{-\frac{\|\boldsymbol{x}_t - e^{-t}\boldsymbol{x}_0\|^2}{2(1 - e^{-2t})}}$, which means

$$\nabla_{\boldsymbol{x}_t} \log p(\boldsymbol{x}_t|\boldsymbol{x}_0) = -\frac{\boldsymbol{x}_t - e^{-t}\boldsymbol{x}_0}{1 - e^{-2t}} = -\frac{\boldsymbol{z}}{\sqrt{1 - e^{-2t}}}.$$

Now we may rewrite $\frac{\partial}{\partial \boldsymbol{\theta}} \mathcal{L}_{CD}^N(\boldsymbol{\theta}, \boldsymbol{\theta}^-)$ as

$$\begin{aligned}
\frac{\partial}{\partial \boldsymbol{\theta}} \mathcal{L}_{CD}^N(\boldsymbol{\theta}, \boldsymbol{\theta}^-) &= \frac{\partial}{\partial \boldsymbol{\theta}} \mathbb{E}[\|\boldsymbol{f}_{\boldsymbol{\theta}}(\boldsymbol{x}_{t_{n+1}}, t_{n+1}) - \boldsymbol{f}_{\boldsymbol{\theta}^-}\left(e^{(t_{n+1} - t_n)}\boldsymbol{x}_{t_{n+1}} + (e^{(t_{n+1} - t_n)} - 1)\nabla \log p_{t_{n+1}}(\boldsymbol{x}_{t_{n+1}}), t_n\right)\|_2^2] \\
&= \frac{\partial}{\partial \boldsymbol{\theta}} \mathbb{E}\left[\|\boldsymbol{f}_{\boldsymbol{\theta}}(\boldsymbol{x}_{t_{n+1}}, t_{n+1})\|_2^2 - 2\left\langle \boldsymbol{f}_{\boldsymbol{\theta}}(\boldsymbol{x}_{t_{n+1}}, t_{n+1}), \boldsymbol{f}_{\boldsymbol{\theta}^-}\left(e^{(t_{n+1} - t_n)}\boldsymbol{x}_{t_{n+1}} + (e^{(t_{n+1} - t_n)} - 1)\nabla \log p_{t_{n+1}}(\boldsymbol{x}_{t_{n+1}}), t_n\right)\right\rangle\right],
\end{aligned}$$

and similarly,

$$\begin{aligned}
\frac{\partial}{\partial \boldsymbol{\theta}} \mathcal{L}_{CT}^N(\boldsymbol{\theta}, \boldsymbol{\theta}^-) &= \frac{\partial}{\partial \boldsymbol{\theta}} \mathbb{E}\left[\left\|\boldsymbol{f}_{\boldsymbol{\theta}}(\boldsymbol{x}_{t_{n+1}}, t_{n+1}) - \boldsymbol{f}_{\boldsymbol{\theta}^-}\left(e^{-t_n}\boldsymbol{x}_0 + \frac{1 - e^{-(t_n + t_{n+1})}}{\sqrt{1 - e^{-2t_{n+1}}}}\boldsymbol{z}, t_n\right)\right\|_2^2\right] \\
&= \frac{\partial}{\partial \boldsymbol{\theta}} \mathbb{E}\left[\|\boldsymbol{f}_{\boldsymbol{\theta}}(\boldsymbol{x}_{t_{n+1}}, t_{n+1})\|_2^2 - 2\left\langle \boldsymbol{f}_{\boldsymbol{\theta}}(\boldsymbol{x}_{t_{n+1}}, t_{n+1}), \boldsymbol{f}_{\boldsymbol{\theta}^-}\left(e^{-t_n}\boldsymbol{x}_0 + \frac{1 - e^{-(t_n + t_{n+1})}}{\sqrt{1 - e^{-2t_{n+1}}}}\boldsymbol{z}, t_n\right)\right\rangle\right],
\end{aligned}$$

thus

$$\begin{aligned}
&\frac{\partial}{\partial \boldsymbol{\theta}} \mathcal{L}_{CD}^N(\boldsymbol{\theta}, \boldsymbol{\theta}^-) - \frac{\partial}{\partial \boldsymbol{\theta}} \mathcal{L}_{CT}^N(\boldsymbol{\theta}, \boldsymbol{\theta}^-) \\
&= 2\mathbb{E}\left[\left\langle \frac{\partial}{\partial \boldsymbol{\theta}} \boldsymbol{f}_{\boldsymbol{\theta}}(\boldsymbol{x}_{t_{n+1}}, t_{n+1}), \right.\right. \\
&\quad \left.\left. \boldsymbol{f}_{\boldsymbol{\theta}^-}\left(e^{-t_n}\boldsymbol{x}_0 + \frac{1 - e^{-(t_n + t_{n+1})}}{\sqrt{1 - e^{-2t_{n+1}}}}\boldsymbol{z}, t_n\right) - \boldsymbol{f}_{\boldsymbol{\theta}^-}\left(e^{(t_{n+1} - t_n)}\boldsymbol{x}_{t_{n+1}} + (e^{(t_{n+1} - t_n)} - 1)\nabla \log p_{t_{n+1}}(\boldsymbol{x}_{t_{n+1}}), t_n\right)\right\rangle\right].
\end{aligned}$$

Notice that

$$
\boldsymbol{f}_{\boldsymbol{\theta}^-}\left(e^{-t_n}\boldsymbol{x}_0 + \frac{1 - e^{-(t_n+t_{n+1})}}{\sqrt{1 - e^{-2t_{n+1}}}}\boldsymbol{z}, t_n\right) - \boldsymbol{f}_{\boldsymbol{\theta}^-}\left(e^{(t_{n+1}-t_n)}\boldsymbol{x}_{t_{n+1}} + (e^{(t_{n+1}-t_n)} - 1)\nabla \log p_{t_{n+1}}(\boldsymbol{x}_{t_{n+1}}), t_n\right)
$$

$$
=\nabla_{\boldsymbol{x}}\boldsymbol{f}_{\boldsymbol{\theta}^-}(e^{(t_{n+1}-t_n)}\boldsymbol{x}_{t_{n+1}} + (e^{(t_{n+1}-t_n)} - 1)\nabla \log p_{t_{n+1}}(\boldsymbol{x}_{t_{n+1}}), t_n)\cdot
$$

$$
\left(\frac{1 - e^{-(t_n+t_{n+1})}}{\sqrt{1 - e^{-2t_{n+1}}}}\boldsymbol{z} - e^{(t_{n+1}-t_n)}\sqrt{1 - e^{-2t_n}}\boldsymbol{z} + (e^{(t_{n+1}-t_n)} - 1)\nabla \log p_{t_{n+1}}(\boldsymbol{x}_{t_{n+1}})\right) + O((t_{n+1} - t_n)^2)
$$

$$
=\nabla_{\boldsymbol{x}}\boldsymbol{f}_{\boldsymbol{\theta}^-}(e^{(t_{n+1}-t_n)}\boldsymbol{x}_{t_{n+1}} + (e^{t_{n+1}-t_n} - 1)\nabla \log p_{t_{n+1}}(\boldsymbol{x}_{t_{n+1}}), t_n)\cdot
$$

$$
(e^{(t_{n+1}-t_n)} - 1)\left(\frac{\boldsymbol{z}}{\sqrt{1 - e^{-2t_{n+1}}}} - \mathbb{E}\left[\frac{\boldsymbol{z}}{\sqrt{1 - e^{-2t_{n+1}}}}\middle| \boldsymbol{x}_{t_{n+1}}\right]\right) + O((t_{n+1} - t_n)^2),
$$

we have (writing the terms related of $\boldsymbol{x}_{t_{n+1}}, t_{n+1}$ with $C(\boldsymbol{x}_{t_{n+1}}, t_{n+1})$ for simplicity)

$$
\frac{\partial}{\partial\boldsymbol{\theta}}\mathcal{L}_{\mathrm{CD}}^N(\boldsymbol{\theta}, \boldsymbol{\theta}^-) - \frac{\partial}{\partial\boldsymbol{\theta}}\mathcal{L}_{\mathrm{CT}}^N(\boldsymbol{\theta}, \boldsymbol{\theta}^-)
$$

$$
=\mathbb{E}[C(\boldsymbol{x}_{t_{n+1}}, t_{n+1})(\boldsymbol{z} - \mathbb{E}[\boldsymbol{z}|\boldsymbol{x}_{t_{n+1}}])] + O((t_{n+1} - t_n)^2)
$$

$$
=\mathbb{E}[C(\boldsymbol{x}_{t_{n+1}}, t_{n+1})\boldsymbol{z}] - \mathbb{E}[\mathbb{E}[C(\boldsymbol{x}_{t_{n+1}}, t_{n+1})\boldsymbol{z}|\boldsymbol{x}_{t_{n+1}}]] + O((t_{n+1} - t_n)^2)
$$

$$
=O((t_{n+1} - t_n)^2)
$$

which finishes the proof. $\qquad\square$

