# OpenReview forum: "Sampling is as easy as keeping the consistency: convergence guarantee for Consistency Models"
_ICLR.cc/2024/Conference — Submitted to ICLR 2024_

### Official Review · Reviewer_CiGA · 2023-10-27

**Soundness:** 3 good
**Presentation:** 3 good
**Contribution:** 3 good
**Rating:** 6
**Confidence:** 2

**Summary:**

This work provides convergence guarantees for the consistency models by Song et al. 2023, which is a one-step generative model achieving state-of-the-art results. The main assumptions are the Lipschitzness of the score function, the score estimation error, and the consistency error. Presented results include consistency mapping error, Wasserstein-2 distance, and TV distance between the mapping and the true probability flow. Multistep consistency sampling is also analyzed to show an improved convergence guarantee compared to the one-step alternative.

**Strengths:**

* The writing is clear and generally good, minus a few typos.
* To my knowledge, this is the first convergence guarantee result for the consistency model, and it is a valuable effort.
* Assumptions on the data distribution are weak.

**Weaknesses:**

* While the assumptions on the data distribution are weak, this work assumes the score estimation error and consistency error are low (Assumption 3, 4), which are major assumptions that are conditioned on the success of the optimization procedure. While it is believable that most usual training procedures can result in low score estimation error since the loss is basically an MSE, it is much harder to reason about the training procedure for the consistency model (8). It is good that the authors acknowledge this point in multiple places.
* Assumption 6 seems unmotivated: the authors attributed it to "technical reason", without further explanation.
* The presentation of the latter results in Section 3.5 seems messy. It looks like the clarity can be improved by using better notations, as many terms are the same.

**Questions:**

* How is $\theta^-$ defined? Is it a moving average of past $\theta$'s?
* Below Assumption 2, it says this paper does not assume convexity or dissipativity on $-\log p$, unlike previous works. What is the reason that the analysis presented here does not require such assumptions?
* What is the significance of obtaining a TV bound, compared to a Wasserstein-2 bound? The result seems a lot more messy compared to the ones concerning W2 errors.
* Minor comments:
  * Some of the $d$'s in (1)(2) are italicized when they shouldn't be
  * In (5), is $dt$ a multiplication of $d$ and $t$?
  * Below (7), "such a mapping exists ..., and is smoothly relied on" What's smoothly relied? And what exactly are the mild conditions mentioned here?
  * Below Assumption 5, "technique reason" -> "technical reason"
  * What does the notation $p P_{OU}^s$ mean, for a Markov kernel $P_{OU}^s$?

---

> ### Author Response · Authors · 2023-11-15
>
> We sincerely thank the reviewer for the constructive comments.
> >While the assumptions on the data distribution are weak, this work assumes the score estimation error and consistency error are low (Assumption 3, 4), which are major assumptions that are conditioned on the success of the optimization procedure. While it is believable that most usual training procedures can result in low score estimation error since the loss is basically an MSE, it is much harder to reason about the training procedure for the consistency model (8). It is good that the authors acknowledge this point in multiple places.
>
> Thanks for the reviewer to point out our deficiency. In the training procedure for the consistency model (8), the parameter $\theta^-$ is a EMA (exponential moving average) to the past $\theta$'s, thus when the trainning converges, we have $\theta^- = \text{stopgrad}(\theta)$, and the Consistency Distillation object will then become a true loss function. If in addition it is small enough (together with good score estimator), which corresponding to assumption 4, our theoretical results shows that the consistency model $f_\theta$ is a good approximator to the true consistency function $f^{ex}$. Besides, Song et al. (2023) also mentioned that, although simply setting $\theta^- = \theta$ will make the Consistency Distillation object be the true loss, the EMA update and "stopgrad" operator can greatly stablize the training procedure and improve the performance.
>
> > Assumption 6 seems unmotivated: the authors attributed it to "technical reason", without further explanation.
>
> Sorry for our unclear expression. Indeed the special time scheduling was firstly suggested by Sitian Chen, *The probability flow ODE is provably fast*, second paragraph in section 3.2 Algorithm. The reason to choose this time scheduling is to control one term of error in Theorem 2: in equation (26), we get the upper bound with a term $\sum_{k=1}^N \frac{h_k^2}{ t_k^{1/2}}$. If we only take a naive time scheduling $h_k \equiv h$, this term now becomes $O(\frac{h^2}{\sqrt{\delta}})$, as $\frac{1}{t_k^{1/2}}$ will become larger when $k \to 0$. One would better choose a relatively smaller discretization step when $t_k$ is small to prevent the discretization error from being unnessary large. Thus a geometrically increasing step size will help a lot.
>
> > The presentation of the latter results in Section 3.5 seems messy. It looks like the clarity can be improved by using better notations, as many terms are the same.
>
> Thanks for your suggestion. We will use better notations to simplify our expression, thus make it more readable.
>
> > How is $\theta^-$ defined? Is it a moving average of past $\theta$'s?
>
> Yes, it is defined as $\theta^- = \text{stopgrad}(\mu\theta^- + (1-\mu) \theta)$. We sorry for putting the definition a little bit later than where it should be, we will modify our expression.
>
> > Below Assumption 2, ... What is the reason that the analysis presented here does not require such assumptions?
>
> In the begining of the theoretical researchs of Diffusion Models, not so many properties of the forward diffusion process, as well as its probability density $p_t(x)$ have been found in Stochastic, PDE anslyses. Thus they need to assume convexity or dissipativity on $\log p_0(x)$ to get prior estimations similar to our Lemma 7, Lemma 11, etc. On the other way, informally the Diffusion Models can be viewed as a Simulated Annealing process, which has better ability to bypass the difficulty from non-convexity or non-dissipativity comparing to traditional Langevin Dynamics.
>
> > What is the significance of obtaining a TV bound, compared to a Wasserstein-2 bound?
> Let the output distribution be $\mu(dx)$. A guarantee of TV error bound can reflect the continuity of its density, thus can further support other operations over Likelihood. However if only have a $W_2$ bound, the output distribution density may be discontinuous, and have some points with infinity density (like Dirac measure).
>
> > In (5), is $dt$ a multiplication of $d$ and $t$?
>
> Yes, it's from the change-of-variable formula for the mapping $x \mapsto e^{-t} x$, where $x$ is a $d-$ dimensional vector.
>
> > Below (7)... What's smoothly relied? And what exactly are the mild conditions mentioned here?
>
> This can be found in many textbooks on Ordinary Differential Equations. A simple case is that when $v$ is differentiable and bounded, then the solution of this ODE uniquely exists, and the mapping through ODE path is differentiable.
>
> > What does the notation $p P^s_{OU}$ mean, for a Markov kernel $P^s_{OU}$?
>
> This is a density defined as follows:
>
> Let $\xi \sim p$, and let the $P^s_{OU}$-corresponded Markov process be started at $\xi$. This Markov process generate another random variable $\xi'$, then we denote the density of $\xi'$ as $p P^s_{OU}$.
>
> > Other Minor comments:
>
> Thanks for the correction and we have fixed the typos.

---

> ### Author Response · Authors · 2023-11-16
>
> We kindly ask the reviewer if they have any outstanding questions or clarifications regarding our paper. We are happy to engage in a dialogue and conduct any additional requested work in the remaining discussion period. Thank you!

---

> > ### Author Response · Authors · 2023-11-17
> >
> > We kindly ask the reviewer if they have any outstanding questions or clarifications regarding our paper. We are happy to engage in a dialogue and conduct any additional requested work in the remaining discussion period. Thank you!

---

> > > ### Comment · Reviewer_CiGA · 2023-11-18
> > > **Reply to authors**
> > >
> > > Thank you for the detailed reply. It has addressed many of my questions. I would like to keep my current score as I agree with other reviewers that the novelty of the analysis can be limited.

---

> > > > ### Author Response · Authors · 2023-11-19
> > > >
> > > > Thank you for your kind response. We will keep finding better ways to describe the theory of consistency models.

---

### Official Review · Reviewer_fM3B · 2023-10-29

**Soundness:** 3 good
**Presentation:** 3 good
**Contribution:** 3 good
**Rating:** 6
**Confidence:** 4

**Summary:**

This paper provides the convergence guarantee for the consisitency models in terms of Wasserstein distance. The authors also show that Multi-step consistency sampling procedure can further reduce the error comparing to one step sampling. Finally, with some Langevin-based modifications, total variation errors are also provided.

**Strengths:**

1. As far as I am concerned, this is the first time a convergence guarantee for consistency models is established.
2. The improvement of multi-step consistency sampling over one step sampling has been clearly demonstrated theoretically.

**Weaknesses:**

1. Not much technical contribution, most of the techniques has already been proposed in the literatures and the proof mostly follows Chen et al.(2023a). Also, Lemma 7 is not new, similar results have been established in [1]
2. There are a lot of typos in the manuscript, e.g., on page 4: equation 4, the expression for $v^{\mathrm{em}}(x,t)$; on page 15: in the second equation, $W_2(\mathcal{N}(0,(1-e^{-2T})I_d), p_T)$ should be $W_2(\mathcal{N}(0,I_d), p_T)$. Please reexamine your manuscript carefully.
3. There are also some technique issues. For example, Lemma 11 in this paper is for the OU noise schedule, while Lemma 1 in Chen et al.(2023a) is for the variance explosion schedule.

[1] Chen, H., Lee, H., and Lu, J. Improved analysis of scorebased generative modeling: User-friendly bounds under minimal smoothness assumptions. arXiv preprint arXiv:2211.01916, 2022a.

**Questions:**

1. In corollary 6, $n_k$ is taken to be a constant $\hat{n}$ for all $k$. While in Song et al. (2023), $n_k$ is suggested to be decreasing. The theoretical results do not seem to support a decreasing $n_k$, any explanation?
2. The results on multistep sampling that only requires a constant lower bound of $T$ is amazing. However, it seems that $L_f$ implicitly depends on $T$, especially when the data distribution is complicated (so it takes more time to transform a simple Gaussian noise to the data distribution). Not sure how pratical this benefit can be.

---

> ### Author Response · Authors · 2023-11-14
>
> We sincerely thank the reviewer for the constructive comments.
> > Not much technical contribution ...
>
> Thanks for your kindly comment. We admit that we use many techniques developed by former researchers, especially in proving Theorem 2, but we still have developed many new proofs, like Theorem 3, Corollary 5, Corollary 6 and Corollary 9. Besides, it is also important to use existing method to solve emerging new problems. We will keep in working harder to develop more new techniques.
>
> > There are a lot of typos in the manuscript ...
> > There are also some technique issues ...
>
> Thanks for the correction. We have fixed these points, especially the Lemma 11 should corresponding to the Lemma 3 in Chen et al.(2023a).
>
> > In corollary 6, $n_k$ is taken to be a constant for all $k$. While in Song et al. (2023), $n_k$ is suggested to be decreasing. The theoretical results do not seem to support a decreasing $n_k$, any explanation?
>
> Actually $n_k$ can be a decreasing sequence and the multistep error can still decrease. Let's look at the recursion inequality in the proof of Corollary 6,
>
> $\mathcal{E}\_k \le L_f e^{-(t_{n_k}-\delta)} \mathcal{E}\_{k-1} + t_{n_k}D,$
>
> the best choice of $t_{n_k}$ should be chosen to minimize the function
>
> $ H(t) = L_f e^{-t - \delta} \mathcal{E}\_{k-1}   + t D$.
>
> However, this is an transcendental equation, and we do not have an explicit evaluation of $\mathcal{E}\_{k-1}$, thus it would be impossible to calculate the best $n_k$. Besides, we can prove that, in upper-bound meaning, the best $n_k$ will finally converge to our defined $\hat n$. Let us suppose the best-choice $n_k$ finally converge to $n^\ast$, with corresponding $W_2$ error
> $\mathcal{E}^\ast$, then we have
>
> $\mathcal{E}^\ast\le L_f e^{-(t_{n^\ast}-\delta)} \mathcal{E}^\ast + t_{n^\ast}D,$
>
> $\mathcal{E}^\ast \le \frac{t_{n^\ast}D}{1-L_f e^{-(t_{n^\ast}-\delta)}}$.
>
> This suggests that we can directly find the extreme point $n^\ast$ of the best-choice sequence $n_k$ by
> $\min_{n} \frac{t_{n}D}{1-L_f e^{-(t_{n}-\delta)}}$, under some prior estimations on $\varepsilon_{cm}, \varepsilon_{sc}, L_f, L_s, d$ and $h$.
>
> Note that a simple fixing $n_k \equiv n^\ast$ also make the $\mathcal{E}\_k$ converging to $\mathcal{E}^\ast$ exponentially, we thus choose to simply use a fixing $n_k$.
>
> > The results on multistep sampling that only requires a constant lower bound of $T$ is amazing. However, it seems that $L_f$
>  implicitly depends on $T$
> , especially when the data distribution is complicated (so it takes more time to transform a simple Gaussian noise to the data distribution). Not sure how pratical this benefit can be.
>
> In actual it can be proven that, for the exact consistency function $f^{ex}$, given the data distribution is bounded-supported, it's Lipschitz constant function $L_f(t)$ has an upper bound irrelavent to $T$. You may look at our rebuttal to Reviewer 8oEy, where we gave a simple proof.
>
> But as the reviewer 8oEy pointed, the Lipschitz constant of $f_\theta$ could still grow with $T$ due to the score error. We thus propose an alternative definition of Lipschitz constant, and an additional penalty to keep the $f_\theta$ having a bounded Lipschitz constant. This can be found in our second rebuttal to Reviewer 8oEy.

---

> ### Author Response · Authors · 2023-11-16
>
> We kindly ask the reviewer if they have any outstanding questions or clarifications regarding our paper. We are happy to engage in a dialogue and conduct any additional requested work in the remaining discussion period. Thank you!

---

> > ### Author Response · Authors · 2023-11-17
> >
> > We kindly ask the reviewer if they have any outstanding questions or clarifications regarding our paper. We are happy to engage in a dialogue and conduct any additional requested work in the remaining discussion period. Thank you!

---

> > > ### Comment · Reviewer_fM3B · 2023-11-19
> > >
> > > Thank you for your response that has addressed many of my questions. I would like to keep my score as the assumption on Lipshitz constant $L_f$ awaits further investigation.

---

> > > > ### Author Response · Authors · 2023-11-19
> > > >
> > > > Thank you for your kind response. We will keep finding better ways to describe the theory of consistency models.

---

### Official Review · Reviewer_8oEy · 2023-10-30

**Soundness:** 4 excellent
**Presentation:** 3 good
**Contribution:** 3 good
**Rating:** 6
**Confidence:** 4

**Summary:**

The paper provides the first convergence guarantee for Consistency Models (CMs), a newly emerging type of one-step generative model with the ability to produce samples comparable to those generated by state-of-the-art Diffusion Models.

**Strengths:**

1. The paper provides the first convergence guarantee for Consistency Models, which is a notable contribution to the field of generative modeling;
2. The results do not rely on strong assumptions about the data distribution, making them broadly applicable to a variety of scenarios;
3. Very clear writing. A comprehensive conclusion on future directions.

**Weaknesses:**

I'm not entirely certain about the reasonability of Assumption 5. See Questions in details.

Besides, there are no major weaknesses. A typo need to be corrected in the revision: at the beginning of section 3.3, it should be 'analyze' instead of 'analysis'.

**Questions:**

The only point I am concerned about is Assumption 5, which assumes the consistency model $f_\theta$ is $L_f$-Lipschitz. But as far as I
 know, even if our model $f_\theta$ can approximate the backward mapping $f^v$ very well, we should still expect that $L_f$ is of $O(e^T)$ (which comes from the Gronwall's Inequality applied on the exact reverse ODE). And so the first term in Corollary 4 could look like $max (d^{1/2}, m)$, which cannot be arbitrarily small. Can you overcome such Gronwall-type of error? If not, I would doubt that the results of this paper prove the efficiency of CMs.

---

> ### Author Response · Authors · 2023-11-13
> **Author Rebuttal**
>
> > The only point I am concerned about is Assumption 5, which assumes the consistency model
>  is $L_f$-Lipschitz. But as far as I know, even if our model can approximate the backward mapping very well, we should still expect that
>  is of $e^T$ (which comes from the Gronwall's Inequality applied on the exact reverse ODE). And so the first term in Corollary 4 could look like $max(d^{1/2},m)$, which cannot be arbitrarily small. Can you overcome such Gronwall-type of error? If not, I would doubt that the results of this paper prove the efficiency of CMs.
>
> Thank the reviewer for pointing this out and we apologize for unsufficient explaination. Your concern is reasonable, and we should prove that the backward ODE would not accumulate the Lipschitz constants exponentially fast. In intuition, this should be true, at least in the ideal case, as when $t \to \infty$, $p_t(x)$ gradually converge to the normal distribution $N(0,I)$, and thus $\nabla \log p_t(x) \to -x$, the particles under the  backward ODE is nealy stopped in large $t$. In fact, we can prove the following statement for the exact backward ODE $dx_t = v(x,t) dt$,
>  where $v(x,t) = -x - \nabla \log p_t(x)$, $p_t(x) = e^{dt}p(e^{t}x) \ast \mathcal{N}(0,(1-e^{-2t})  I_d) \sim \int_{\mathbb{R}^d} e^{dt} p(e^{t}u)e^{-\frac{\Vert x-u\Vert^2}{2(1-e^{-2t})}}du$ with a bounded data distribution density $p_0(x) = p_{data}(x), \text{supp } p(x) \subseteq B(0,R)$ for some $R > 0$.
>
> ***
>
> **Property** . The velocity field $v(x,t)$ in the exact backward ODE satisfies:
>
> $ \Vert \nabla v(x,t) \Vert_{op} \le e^{-2t} M,$
> for some $M > 0$.
>
> *Proof*.
> By the proof of Lemma 7 in our work, we have proved that
>
> $\nabla^2 \log (\mu \ast \psi_{\sigma^2}(x)) = \frac{1}{\sigma^4} \int_{\mathbb{R}^d} (u -\mathbb{E}\_{\mu_{x,\sigma^2}}[u] )\otimes (u -\mathbb{E}\_{\mu_{x,\sigma^2}}[u] )  \mu_{x, \sigma^2}(du) - \frac{1}{\sigma^2} I_d$
>
> where
>
>  $\psi_{\sigma^2}(u - x) \sim e^{-\frac{\Vert x - u\Vert_2^2}{2\sigma^2}},$
>
> $  \mu_{x, \sigma^2}(du) = \frac{e^{-\frac{\Vert x - u\Vert_2^2}{2\sigma^2} }\mu(du)}{\int_{\mathbb{R}^d} e^{-\frac{\Vert x - \tilde u\Vert_2^2}{2\sigma^2}} \mu(d\tilde u)}.$
>
> Taking $\mu(u) = e^{dt} p_0(e^t u)$ which is bounded on a set of radius $e^{-t}R$ and $\sigma^2 = 1-e^{-2t}$, noticing that
> $\nabla v(x,t) = -I_d - \nabla^2 \log (\mu \ast \psi_{\sigma^2})$, we immediately get the result.
>
> ***
>
> Now we can prove that the analytical consistency mapping $f^v(x,t)$ has a uniform Lipschitz constant for all $t \ge \delta$. We notice that, for two different end points $x_T^1, x_T^2$, we define
>
> $d x^1_t = v(x_t^1,t) dt$
>
> $d x^2_t = v(x_t^2,t) dt$
>
> and
> $f^v(x_T^1,T) - f^v(x_T^2,T) = x^1_\delta - x^2_\delta$
>
> with early stopping time $\delta >0$. Notice that,
>
> $ \frac{d}{dt} \Vert x^1_t - x^2_t \Vert^2_2 = \langle x^1_t - x^2_t, v(x_t^1 ,t) - v(x_t^2 , t)\rangle \le \max  \Vert \nabla v(x,t) \Vert_{op} \Vert x^1_t - x^2_t \Vert_2^2 = e^{-2t} M \Vert x^1_t - x^2_t \Vert^2_2,$
>
> thus by Gronwall's Inequality,
>
> $ \Vert x^1_\delta - x^2_\delta \Vert^2_2 \le   \Vert x^1_T - x^2_T \Vert^2_2  \exp( \int_{\delta}^{T}e^{-2t}M) \le  \Vert x^1_T - x^2_T \Vert^2_2  \exp( \frac{M}{2}(1- e^{-2T})),$
>
> which shows a bounded Lipschitz constant when $T \to \infty$.
>
> When dealing with the model $f^{em}$ under the approximated score function $s_\phi$, it would be much more difficult, as we should ensure $-x - s_\phi(x,t)$ has a similar operator bound like $-x - \nabla \log p_t(x)$. This can be proved by further strengthen the assumption 3 to a $L_\infty$ norm, and multiply the right-hand-side with an exponential-decreasing term $e^{-2t_n}$,
>
>  $\Vert s_{\phi}(x,t_n) - \nabla \log p_{t_n}(x) \Vert_2^2 \le \frac{e^{-2t_n}}{1-e^{-2t_n}}\varepsilon^2_{sc}$ for all $x$ and $t_n$.
>
> In words, we believe that our assumption 5 is reasonable, however further analysis can be conducted to replace this assumption to more reasonable assunptions. This will be our future work.

---

> > ### Comment · Reviewer_8oEy · 2023-11-13
> >
> > Thanks for your response. I agree with your argument here. But does it mean that we need an $L^\infty$ score estimation assumption instead an $L^2$ one? If so, I still cannot see why assumption 5 is reasonable. You are kind of skipping the Gronwall type of error induced by score estimation and discretization.

---

> ### Author Response · Authors · 2023-11-14
>
> Thank you for your kindly response.
>
> You are correct, as our existing assumptions can not ensure the backward $f_\theta$ have a uniform Lipschitz constant. Actually in our first response we said we need an $L^\infty$ score, that should still not enough: we need further ensure $||\nabla s_\phi(x,t) - \nabla^2 p_t(x)||$ to be small, such that the Gronwall type of error introduced by score estimation won't increase exponentially. This can be partially reduced by the so-called higher order score matching ([1], Theorem 4.1), however still not enough as the higher order score matching only guarantee a small $L^2$ error either. This is just why the theory and performance for probability ODEs are weaker than probability SDEs.
>
> Nevertheless, as the consistency model $f_\theta$ offers us a direct way to estimate the endpoints of the ODE flows, comparing to traditional higher order score matching for ODE flows that carefully restrict the score function, we can directly constrain $f_\theta$ to satisfy some Lipschitz conditions. This is reasonable, as the true consistency mapping $f^{ex}$ should have this property. Here I'll explain how to introduce this constraint inside the trainning.
>
> First let us modify our Assumption 5, as we do not need global and strong Lipschitz conditions in $L^\infty$ meaning; actually the only usage of assumption 5 is in Equation (20), which is local and under $L^2$ meanings.
>
> **Assumption 5'**
> The consistency model $f_{\theta}(x,t_n)$ satisfies
>
> $ \mathbb{E}\_{x_{t_{n+1}} \sim p_{t_{n+1}} }\left[\left\Vert   f_\theta( \hat x_{t_n}^\phi,t_n) - f_\theta( x_{t_n},t_n)    \right\Vert_2^2 \right] \le L_f^2     \mathbb{E}\_{x_{t_{n+1}} \sim p_{t_n+1} }\left[\Vert \hat x_{t_n}^\phi - x_{t_n}  \Vert_2^2\right] $
> for all $n \in [\\![1,N-1]\\!]$.
>
> There are many methods to restrict our model to satisfy this assumption. For example, we can add a penalty term
>
> $ \gamma \max\left(0, \mathbb{E}\_{x_{t_{n+1}} \sim p_{t_n+1} }\left[\left\Vert   f_\theta( \hat x_{t_n}^\phi,t_n) - f_\theta( x_{t_n},t_n)    \right\Vert_2^2 - L_f^2    \Vert \hat x_{t_n}^\phi - x_{t_n}  \Vert_2^2\right]\right),$
>
> where $\gamma$ is a scaling parameter. We should choose $L_f \ge \exp(\frac{M}{4})$, where $M$ is the operator norm of exact backward ODE field $v(x,t) = -x - \nabla \log p_t(x)$ introduced in the discussion before. We can approximate $x_{t_n}$ using initial point $x_{t_{n+1}}$ and finer stepsize ODE solvers as a reference. We can choose less sample points than the main Consistency Loss term to save computational costs.  Although this is not exactly what it should be, we believe this can preventing the consistency models from getting an exponentially-increasing Lipschitz constants.
>
> ***
>
>
> [1] Maximum Likelihood Training for Score-Based Diffusion ODEs by High-Order Denoising Score Matching

---

> > ### Author Response · Authors · 2023-11-16
> >
> > We kindly ask the reviewer if they have any outstanding questions or clarifications regarding our paper. We are happy to engage in a dialogue and conduct any additional requested work in the remaining discussion period. Thank you!

---

> > > ### Author Response · Authors · 2023-11-17
> > >
> > > We kindly ask the reviewer if they have any outstanding questions or clarifications regarding our paper. We are happy to engage in a dialogue and conduct any additional requested work in the remaining discussion period. Thank you!

---

> > ### Comment · Reviewer_8oEy · 2023-11-18
> >
> > Thank you for your response. I agree with your argument here. Although I am still not sure if restricting the model to satiscy the Lipschitz assumption is reasonable either in theory/practice, I will change my score.

---

> > > ### Author Response · Authors · 2023-11-19
> > >
> > > Thank you for your kind response. We will keep finding better ways to describe the theory of consistency models.

---

### Official Review · Reviewer_N63V · 2023-11-10

**Soundness:** 3 good
**Presentation:** 3 good
**Contribution:** 3 good
**Rating:** 8
**Confidence:** 3

**Summary:**

The authors of this work provide the first convergence guarantees for Consistency Models, a new class of diffusion models that achieves one-step generation. Under classical assumptions (smooth data density, bounded second moment of the data distribution, and an L2 bound on the score estimation) plus an additional assumption on the error of the consistency sampler, the authors provide a bound on the Wasserstein that is polynomial in all problem parameters. The authors further analyze multistep consistency sampling which removes the linear dependence on the total diffusion time $T$. Finally, the authors provide TV bounds by: i) early stopping and ii) modifying the reverse process with Langevin correctors.

**Strengths:**

- The authors provide the first analysis for Consistency Diffusion Models, which is an emerging class of diffusion models with nice theoretical and practical properties. It is nice to see that the theory follows closely the practical advancements and complements our understanding of why certain learning algorithms work.
- The authors do a thorough analysis of Consistency Diffusion Models, providing bounds for Wasserstein distance (for smooth and bounded densities) and TV distance (under certain modifications).
- The obtained bounds match the ones known for Score-Based Generative Models from the "Sampling is as easy as learning the score" paper.
- The theoretical analysis is novel, especially for the result of Theorem 2.

**Weaknesses:**

Even though I overall consider this a strong submission, there are some weaknesses that need to be addressed or clarified.


- The presentation of the paper could be improved. There are several typos in the main text -- I would encourage the authors to do another pass over the manuscript and fix them. Some examples:
     * "be further weaken" -> "be further weakened"
     * "consistency model is nature" -> "is natural"
     * same sentence, fix formatting of the parenthesis.
     * Figure 1: punctuation missing.
     * "a asymptotic analysis" -> "an asymptotic analysis"
     * "For technique reason" -> for technical reasons. It would also help to explain what these reasons are.
- To improve the clarify of the paper, I think it would be better to explain what $\theta_{-}$ is the first time it is introduced.
- For Assumption 4, I think that the expectation should be over $x_{t_{n+1}}$. Given a $x_{t_{n+1}}$ everything is deterministic. Does that affect the proof of Theorem 2?
- Corollary 4 is a little confusing. It seems that we require a lower bound on $T$, however, in the text, it is presented as having a large value of $T$ is a bad thing -- which is why the authors claim that multistep consistency sampling helps. I think the way this result is presented is a little misleading since it requires the learning errors to be O(1/T).
- I am a little concerned about how much of the complexity of the problem is hidden in the upper-bound assumption on the learning of the consistency model. The model is trained to solve the whole probability flow ODE. If the score is inaccurately learned, the prediction of the solution can be poor because of error propagation. It seems that the error in the learning of the score and the consistency errors are actually related.
- I also don't understand why this special time scheduling is needed. I don't think this is the scheduling that has been used in practice. I also don't recall seeing this assumption in prior work and it would be great to understand where this is coming from.

**Questions:**

See weaknesses above. Repeating here the points for completeness:
- For Assumption 4, I think that the expectation should be over $x_{t_{n+1}}$. Given a $x_{t_{n+1}}$ everything is deterministic. Does that affect the proof of Theorem 2?
- Corollary 4 is a little confusing. It seems that we require a lower bound on $T$, however, in the text, it is presented as having a large value of $T$ is a bad thing -- which is why the authors claim that multistep consistency sampling helps. I think the way this result is presented is a little misleading since it requires the learning errors to be O(1/T).
- I am a little concerned about how much of the complexity of the problem is hidden in the upper-bound assumption on the learning of the consistency model. The model is trained to solve the whole probability flow ODE. If the score is inaccurately learned, the prediction of the solution can be poor because of error propagation. It seems that the error in the learning of the score and the consistency errors are actually related.
- I also don't understand why this special time scheduling is needed. I don't think this is the scheduling that has been used in practice. I also don't recall seeing this assumption in prior work and it would be great to understand where this is coming from.

---

> ### Author Response · Authors · 2023-11-13
> **Author Rebuttal**
>
> We sincerely thank the reviewer for the constructive comments.
> > The presentation of the paper could be improved.
> > To improve the clarify of the paper, I think it would be better to explain what $\theta^{-}$ is the first time it is introduced.
>
> Thanks for pointing this out, we have modified all the mentioned typos or misleading in the updated paper
>
> > For Assumption 4, I think that the expectation should be over $x_{t_{n+1}}$. Given a $x_{t_{n+1}}$  everything is deterministic. Does that affect the proof of Theorem 2?
>
> Thanks for pointing this out, we have modified all the mentioned typos in the updated paper. This would not affect the proof of Theorem 2, as in each individual term that needs to calculate expectations, $x_t$ are keeped in the same ODE path for all $t\in[\delta, T]$.
>
> > Corollary 4 is a little confusing. It seems that we require a lower bound on $T$, however, in the text, it is presented as having a large value of $T$ is a bad thing -- which is why the authors claim that multistep consistency sampling helps. I think the way this result is presented is a little misleading since it requires the learning errors to be $O(1/T)$.
>
> That's correct: as $T$ increase, the discretization error will increase; however, if $T$ is not large enough, the error introduced by replacing $p_T$ to a normal distribution would become large. This is also obtained in other works such as Theorem 2, *Sampling is as easy as learning the score: theory for diffusion models with minimal data assumptions*.
>
> > I am a little concerned about how much of the complexity of the problem is hidden in the upper-bound assumption on the learning of the consistency model. The model is trained to solve the whole probability flow ODE. If the score is inaccurately learned, the prediction of the solution can be poor because of error propagation. It seems that the error in the learning of the score and the consistency errors are actually related.
>
> The definition of consistency error can be disentangle to the score matching error, as our Assumption 4 do not need $\hat x_{t_n}^\phi$ to be exact enough: in fact, given any score model $s_\phi(x,t)$, no matter how far from the exact score $\nabla \log p_t(x)$,  we can always define the consistency mapping $f^{em}$. And if one can not learn the score well enough, one still can not generate good enough samples by diffusion models, let alone consistency models (in consistency distillation way).
>
> > I also don't understand why this special time scheduling is needed. I don't think this is the scheduling that has been used in practice. I also don't recall seeing this assumption in prior work and it would be great to understand where this is coming from.
>
> Sorry for the insufficient explaination. The special time scheduling was firstly suggested by Sitian Chen, *The probability flow ODE is provably fast*, second paragraph in section 3.2 Algorithm. The reason to choose this time scheduling is to control one term of error in Theorem 2: in equation (26), we get the upper bound with a term $\sum_{k=1}^N \frac{h_k^2}{ t_k^{1/2}}$. If we only take a naive time scheduling $h_k \equiv h$, this term now becomes $O(\frac{h^2}{\sqrt{\delta}})$, as $\frac{1}{t_k^{1/2}}$ will become larger when $k \to 0$. One would better choose a relatively smaller discretization step when $t_k$ is small to prevent the discretization error from being unnessary large. Thus a geometrically increasing step size will help a lot.

---

> ### Author Response · Authors · 2023-11-16
>
> We kindly ask the reviewer if they have any outstanding questions or clarifications regarding our paper. We are happy to engage in a dialogue and conduct any additional requested work in the remaining discussion period. Thank you!

---

> > ### Author Response · Authors · 2023-11-17
> >
> > We kindly ask the reviewer if they have any outstanding questions or clarifications regarding our paper. We are happy to engage in a dialogue and conduct any additional requested work in the remaining discussion period. Thank you!

---

### Meta-Review · Area_Chair_5JAs · 2023-12-03

**Metareview:**

The authors give the first convergence guarantees for consistency models (a type of 1-step generative models): given assumptions on score-matching and consistency errors and smoothness, CM's generate samples in 1 step with small $W_2$ error. They follow the framework for error analysis for diffusion models. They further show multi-step consistency sampling can reduce the error, and (similar to related work) also show TV bounds by early stopping and adding a Langevin corrector.

The main strength is that these are the first convergence guarantees for these models, and under the given assumptions, they match what is known for diffusion models. The main strength of CM's over DM's is 1-step generation.

The main weakness is an issue independently raised by several reviewers: In Assumption 5, the Lipschitz constant $L_f$ of the consistency map can be large.
While the authors gave some derivations of Lipschitzness in the response, it seems that this is a subtle issue and needs to be more carefully evaluated. I would suggest that the paper be revised to include a careful discussion of this issue.
In particular, there is an exponential dependence on an unspecified M. There are simple cases where $L_f$ can be expected to be large (e.g., a mixture of 2 separated gaussians; the Lipschitz constant from mapping a single gaussian could intuitively be large). More careful work needs to be done to understand to what extent this is an issue, and to clearly delimit the class of distributions for which $L_f$ can be expected to be small/large.

**Justification For Why Not Higher Score:**

The paper is missing more discussion of the limitations of Assumption 5.

**Justification For Why Not Lower Score:**

N/A

---

### Decision · Program_Chairs · 2024-01-16

Reject